# Changes in social norms during the early stages of the COVID-19 pandemic across 43 countries

The emergence of COVID-19 dramatically changed social behavior across societies and contexts. Here we study whether social norms also changed. Specifically, we study this question for cultural tightness (the degree to which societies generally have strong norms), specific social norms (e.g. stealing, hand washing), and norms about enforcement, using survey data from 30,431 respondents in 43 countries recorded before and in the early stages following the emergence of COVID-19. Using variation in disease intensity, we shed light on the mechanisms predicting changes in social norm measures. We find evidence that, after the emergence of the COVID-19 pandemic, hand washing norms increased while tightness and punishing frequency slightly decreased but observe no evidence for a robust change in most other norms. Thus, at least in the short term, our findings suggest that cultures are largely stable to pandemic threats except in those norms, hand washing in this case, that are perceived to be directly relevant to dealing with the collective threat.

Societies vary extensively in the kinds and number of social norms—the unwritten social rules that guide behavior[1,2]—that they adopt and the extent to which people within those societies follow them. From religious ceremonies and dress codes to environmental conservation and infection-containment, we embrace an astonishing diversity of social norms. An influential theory proposes that societies with many strong social norms, and in which individuals who deviate from the script face severe social punishment, can be classified as tight, while those that are permissive, have few and weak social norms, and norm-breakers are subject to little punishment are known as loose[3,4]. Such differences in cultural tightness are also reflected in prevailing socio-political institutions and practices. Tighter countries, or regions, are likelier to have restrictive socio-political institutions (e.g., government, media, education, legal, and religious), stricter constraints across everyday situations (e.g., public park, library, restaurant, workplace, classroom), more incremental innovation, lower debt, and stronger metanorms (norms about punishment) among others[3,5–11]. Loose cultures are instead more open to new ideas, more predisposed to change and substantial innovation, but may have difficulties in facing collective risks. Indeed, recent work finds that looser societies had less success in limiting COVID-19 cases and deaths in the first stages of the pandemic[12].

Given the broad practical and scientific importance of tightness-looseness, it is essential to understand what factors are associated with these differences across countries and cultures. Tightness-Looseness theory[3] contends that societies that have experienced chronic ecological and social threats—frequent disease, warfare, and environmental catastrophes—throughout history develop tighter cultures to maintain order and survive chaos and crises. In contrast, societies with less exposure to such ecological threats can afford to develop looser cultures that allow innovation and creativity at the cost of order. This core hypothesis, that social norm strength is related to the threats that nations have (or have not) historically encountered, is well supported by correlational evidence from cross-sectional surveys[3,6,7], ethnographic datasets[8], a long-term online experiment[13], and a long-term survey about social distancing norms[14]. Moreover, computational models have shown that dramatic increases in threat cause tightening[15]. On the other hand, cultural evolution has been argued to be a slow process[16,17], suggesting the alternative that norm strength is stable after a collective threat. The COVID-19 pandemic provides an opportunity to examine whether tightening naturally occurs or if culture remains stable in the early stages of a collective threat. This knowledge can help us not only predict the future responses of

✉ e-mail: giulia.andrighetto@istc.cnr.it

countries to similar situations and potentially identify effective interventions to deal with these crises but also to better anticipate social changes that can impact our societies for generations to come.

Here we address this question by studying a dataset on cultural tightness, social norms, and metanorms—norms about the punishment of norm-breakers[18]—and exploit variation in disease severity due to the COVID-19 pandemic to test whether tightening evolves after a collective threat. Specifically, we combine data from a survey collected between April–December 2019 (Wave 1)[5] prior to the pandemic with a repeat of the same survey, in the same countries and sampled from the same populations, that we conducted in March–July 2020 (Wave 2) during the first months of the COVID-19 pandemic. The combined data come from 30,431 respondents (samples from both students and non-students) and cover 55 cities in 43 countries (see Table S1 for summary).

The follow-up data (Wave 2) were collected during the initial stages of the pandemic so they capture the early changes (or their stability) in norms that occurred. While this means that we cannot infer the long-run consequences of the pandemic on norms, it also presents important advantages. First, our data provide an insight into norm change under extreme circumstances—while social, political, and economic systems were in upheaval—which provides strong stimuli for change to occur potentially shaping norms. Put differently, if norm change occurs, then there is a good chance we should be able to observe this in the early stages. Second, early data give an insight into the non-equilibrium dynamics of how cultures move from one stable state to another. Third, we are able to test the boundaries of tightness-looseness theory in terms of timeline: our data indicate a lower bound on the time that may be needed for large-scale norm change to occur in response to pandemic threat. Fourth, endogeneity issues are reduced. Specifically, it reduces the possibility for other large-scale shocks to affect the data and the possibility of time varying factors (e.g. hospital infrastructure development) to confound our results.

To study whether a change in disease threat is associated with a change in norms, we study five outcomes. (i) Tightness-looseness: elicited using the standard six questions (e.g., "There are many social norms that people are supposed to abide by in this country") with ratings standardized to control for response sets[3,5]. (ii) Situation-specific social norms' strength: measured with disapproval of norm-breaking in four settings (e.g., listening to music on headphones at a funeral[19]) and stealing shared resources[20]. (iii) Metanorm strength: for each of the prior scenarios respondents also rated the appropriateness of different responses to the norm-breaker by another individual (verbal confrontation, ostracism, gossip, physical punishment, and non-action)[5,18]. (iv) Frequency of punishing norm-breakers. (v) Hand washing norms: respondents indicated the situations (e.g., after shaking someone's hand) in which people should wash their hands. Our core expectation is that these outcomes are higher after the emergence of COVID-19 than before.

These outcomes vary in their relevance to the COVID-19 pandemic. Hand hygiene is strongly related, stealing is partly related (i.e. stealing shared resources during a pandemic is particularly harmful), while others, such as listening to music on headphones at a funeral, are unrelated to the pandemic. Intuitively, norms most related to preventing disease spread should change the most. Yet tightness-looseness theory does not make such detailed predictions. Instead, it proposes the overarching hypothesis that norms and metanorms strengthen. Such a broad change may happen for two interlinked reasons: in the presence of threats, people rely more on social norms as heuristics to safely determine what to do and this increase in conformity leads to a general tightening[21]; it is beneficial to have tight norms across the board since tightening even irrelevant norms can increase a general norm-following tendency that implies increased norm-following for the more relevant ones.

To gain a deeper insight into the mechanisms that may be associated with change, we exploit the heterogeneity across countries in their exposure to COVID-19 and we collected data on three pathways through which we conjecture that COVID-19 pandemics may shape norms. Two of these are the respondent's beliefs about the prevalence of COVID-19 and their fear of COVID-19, as we conjecture that disease threat shapes the strength of norms through individuals' perceptions. The final pathway concerns government policy. By implementing strict (or lenient) anti-disease policies, governments can signal to their citizens the severity of the threat. Moreover, they impose policies that change their citizens' behavioral patterns (e.g., not shaking hands, socially isolating) and these may have consequences on social expectations and norms. While all countries in the sample have been exposed to the pandemic, the continuous variation in our collected measures helps shed light on the association between cultural change and intensity of COVID-19 pandemic. The study, including the hypotheses and analyses, was pre-registered with the Open Science Framework (see Methods).

Overall, we find that in the short term, the global threat posed by the COVID-19 pandemic was associated with a significant strengthening of social norms related to hand washing, a behavior highly relevant to limit disease spread. Contrary to our initial predictions, other established social norms governing our daily lives exhibit resilience and remain largely unchanged. In addition, cultural tightness slightly decreased accompanied by small decrease in punishment frequency. These findings suggest that the immediate impact of a global threat is selective in changing those norms that are directly relevant to cope with the threat and emphasizes the adaptive nature of societies in the face of a collective crisis.

## Results

Our analytic strategy proceeds in two stages. We first compare Wave 1 to Wave 2 averages using multilevel models with individual responses grouped on city and country. We then seek to identify the mechanisms associated with changes for only those outcomes that show significant associations which are robust across both models and sub-items. To do this we use the change across waves (Wave 2 - Wave 1) as the dependent variable as predicted by perceived prevalence, fear, and government stringency and use country-level observations and OLS regression models with heteroskedastic robust standard errors. Prevalence is measured using "What percent of people living in your province do you think have been infected with COVID-19?" and fear is the combination of three items (Cronbach's $\alpha = 0.84$, see Methods for country-level variation). To capture variation in governmental policies, we use the Stringency Index from the Oxford COVID-19 Government Response Tracker[22]. This second stage of our analysis is similar in spirit to a difference-in-differences design but differs to the classical setup in that we have no entirely untreated control group—all countries in our sample were to some extent affected by the emergence of the COVID-19 pandemic—and instead of a treated and untreated group, we have many groups with different COVID-19 pandemic exposure levels. All analyses account for age, gender, and student status to control for any sample composition differences between the waves (see Methods). We also check whether deaths and cases, which account for the different levels of COVID-19 across countries, affect our results and find that they do not (see Supplementary Materials).

After our analyses were conducted, we added equivalence tests using the two one-sided tests procedure[23–25] to identify whether significant changes that we find are practically meaningful and if non-significant findings provide evidence for the absence of a meaningful change. In this procedure, we specify a series of smallest effect size of interest (SESOI) and then compare Wave 1 to Wave 2 changes and the mechanism associations to these SESOIs. Our SESOIs were set ex-post and not pre-registered and, given the lack of existing literature, or even data, concerning the changes in our outcome variables, there is large

uncertainty about how the SESOI should be set (see Methods for discussion). Consequently, we use a benchmark-based approach and set the SESOI to Cohen's $d = 0.1$ (a small effect size[26]) for our main individual-level analyses and $\beta = \pm 0.10$ (a small effect size[26]) for the mechanisms analyses (see Methods for details).

## Tightness-Looseness

Tightness decreases ($\bar{x}_1 = 1.90$, $\bar{x}_2 = 1.81$; Fig. 1A; Table S1) although the effect size is small (Cohen's $d = 0.11$; $b = -0.028$, 95% CI = [−0.047; −0.009], $p = 0.003$; Table S2), and the change is heterogeneous across countries (varying slope model: $b = -0.037$, 95% CI = [−0.073; −0.001], $p = 0.042$; random effect variance $\tau_{11} = 0.01$; Table S2; Figure S2). In most countries, the change is not significant (81.4%; 35/43), it is negative in 16.3% (7/43) and even positive in 2.3% (1/43) (Fig. S2). Countries that have higher fear levels towards COVID-19 reduced their tightness the most ($b = -0.081$, 95% CI = [−0.157; −0.005], $p = 0.037$; Table S3) though this association is small. Perceived prevalence and government stringency are not significantly associated with change in tightness-looseness ($b = -0.003$, 95% CI = [−0.010; 0.003], $p = 0.306$ and $b = 0.0003$, 95% CI = [−0.002; 0.001], $p = 0.721$, respectively; Table S3).

## Situation-specific norms

Situation-specific norm strength decrease slightly from Wave 1 to Wave 2 ($\bar{x}_1 = 1.15$, $\bar{x}_2 = 1.12$; Fig. 1B; Cohen's $d = 0.04$; $b = -0.017$, 95% CI = [−0.028; −0.006], $p = 0.003$; Table S4) but this is not robust as it becomes non-significant when allowing for heterogeneous effects across countries (varying slope model: $b = -0.011$, 95% CI = [−0.054; 0.033], $p = 0.628$, $\tau_{11} = 0.02$; Table S4; Fig. S3). Analyses conducted on the five specific norm-breaking scenarios separately also show no consistent pattern (three are negative and two are positive) and the size of the changes is minimal (Table S5). These results demonstrate that COVID-19 has no consistent effect on situation-specific norms, and, even where it does, the effect is minor.

## Metanorms

We report similar findings for metanorms (Fig. 1C). There is no significant change across the waves ($\bar{x}_1 = 2.15$, $\bar{x}_2 = 2.17$; Cohen's $d = 0.03$; $b = 0.006$, 95% CI = [−0.001; 0.013], $p = 0.120$; Table S6; Fig. S4) and there is little consistency across the different kinds of punishments: approval of ostracism slightly increases ($b = 0.028$, 95% CI = [0.015; 0.040], $p < 0.001$; Table S7) while gossip approval slightly decreases ($b = -0.024$, 95% CI = [−0.035; −0.013], $p < 0.001$; Table S7). Estimates from our models show no significant change in verbal confrontation, physical confrontation, and non-action (reverse coded) items.

## Punishing frequency

In contrast, we find a statistically significant decrease in frequency of punishment ($\bar{x}_1 = 3.00$, $\bar{x}_2 = 2.96$; Fig. 1D; Cohen's $d = -0.07$; $b = -0.034$, 95% CI = [−0.047; −0.022], $p < 0.001$; Table S8). This effect remains

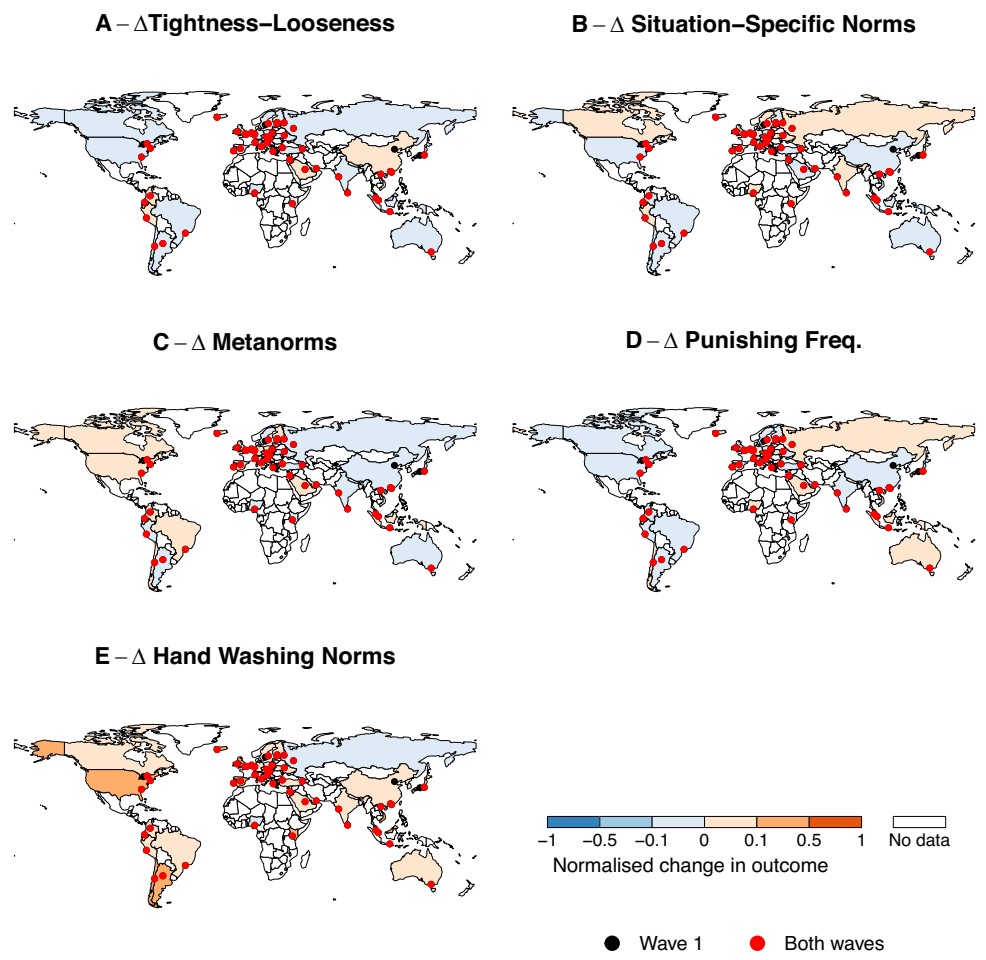

**A – Δ Tightness–Looseness**

**B – Δ Situation–Specific Norms**

**C – Δ Metanorms**

**D – Δ Punishing Freq.**

**E – Δ Hand Washing Norms**

−1   −0.5   −0.1   0   0.1   0.5   1   No data
Normalised change in outcome

● Wave 1      ● Both waves

**Fig. 1 | Changes in outcomes (Wave 2 - Wave 1).** (**A**) tightness-looseness, (**B**) situation-specific norms, (**C**) metanorms, (**D**) punishing frequency and (**E**) hand washing norms. Tightness and punishing frequency slightly decrease while hand washing norms increase after the emergence of the COVID−19 pandemic. Other social and metanorms display non-robust changes. Change in appropriateness items is computed by scaling the average change in each country to the maximum possible change. Hence, the index can take values from −1 to +1. Red and black dots depict sampled cities; red dots represent cities sampled in both waves while black dots refer to cities only sampled in Wave 2. Indonesia is not included in hand washing norm data because of a mistake in the survey translation (see Methods).

negative and significant with a varying slopes model ($b = -0.031$, 95% CI = [$-0.059$; $-0.003$], $p = 0.028$, $\tau_{11} = 0.01$; Table S8) and it is generally consistent across sub-items with the frequency of gossip ($b = -0.091$, 95% CI = [$-0.112$; $-0.070$], $p < 0.001$; Table S9) and confronting ($b = -0.021$, 95% CI = [$-0.041$; $-0.002$], $p = 0.035$; Table S9) both decreasing. Perhaps due to distancing and self-isolating measures, avoiding shows no significant change ($b = 0.011$, 95% CI = [$-0.012$; $0.034$], $p = 0.335$; Table S9). Frequency of gossiping tended to decrease more in countries with a higher level of fear of COVID-19 ($b = -0.139$, 95% CI = [$-0.261$; $-0.016$], $p = 0.028$; Table S10). The other change in punishing frequency categories, including the overall index, are not associated with the mechanism variables (Table S10).

### Hand washing norms

Hand washing norms increase on average ($\bar{x}_1 = 4.07$, $\bar{x}_2 = 4.50$; Fig. 1E; Cohen's $d = 0.32$; $b = 0.420$, 95% CI = [$0.390$; $0.450$], $p < 0.001$; Table S11) and in almost every country (41 out of 42 countries, Fig. 1E; all countries when considering only COVID relevant items, Fig. S1). Results remain unchanged when accounting for country-level heterogeneity (varying slope model: $b = 0.433$, 95% CI = [$0.361$; $0.506$], $p < 0.001$; $\tau_{11} = 0.04$; Table S11 Fig. S3). The increase is most strongly associated in the categories perceived to be relevant to reducing COVID-19 spread (Table S12). Fear of COVID-19 accounts for most of the increase across all items ($b = 0.040$, 95% CI = [$0.004$; $0.076$], $p = 0.032$; Table S13) and this effect becomes stronger when predicting only the change of COVID-relevant items ($b = 0.092$, 95% CI = [$0.035$; $0.148$], $p = 0.002$; Table S13). Perceived prevalence does not predict hand washing norm change both when considering all items ($b = 0.002$, 95% CI = [$-0.0003$; $0.0049$], $p = 0.085$; Table S13) and relevant items ($b = 0.004$, 95% CI = [$-0.001$; $0.008$], $p = 0.086$; Table S13) but does so after shaking hands ($b = 0.004$, 95% CI = [$0.001$; $0.008$], $p = 0.015$; Table S13). Governmental stringency does not predict change in hand washing norms ($b = 0.0002$, 95% CI = [$-0.001$; $0.001$], $p = 0.723$; Table S13).

### Equivalence tests

For tightness-looseness, situation-specific norms, metanorms, and punishing frequency, we find that the between wave variation observed are statistically equivalent (all $p < 0.001$) implying that the differences are statistically smaller than the SESOI we set. For hand washing norms, we find that the change is statistically greater than the SESOI, exceeding the upper equivalence bound (see Methods for details). For the mechanisms analyses, fear of COVID-19 is significantly associated with the outcomes of tightness-looseness and hand washing norms while all the other relevant mechanism coefficients are not significantly different to zero. Yet they all overlap with either the upper or lower equivalence bounds meaning that there is insufficient evidence to conclude a negligible effect (see Methods for details).

## Discussion

Our findings show that even a crisis as profound, global, and multifaceted as COVID-19 does not dramatically change the social norms of cultures in the short-term, except those believed to directly reduce disease spread, hand washing norms in this case. Nevertheless, and contrary to our expectations, we find a small decrease in tightness and punishing frequency and no significant robust changes in most social norms and metanorms in the early stages of the pandemic. Importantly, the non-significant findings are due to the absence of substantial changes and not because of a lack of power. What explains these results? One possibility is that the key prediction of tightness-looseness theory needs to be revised. Due to existing large-scale studies across multiple fields, which support the association between threat and tightness-looseness[3,6–12] and more broadly social norm strength[13,27,28], we do not think this is the likeliest explanation. Instead, we think that there are more probable interpretations.

A distinct possibility is that cultural evolution is slow and extensive time is necessary between a collective threat and a subsequent change in cultures[16,17]. Indeed, if cultures do change slowly, we may expect specific cultural evolutionary mismatches—i.e., when traits that evolved in one environment become disadvantageous in a different environment[29,30]. Specifically, tight societies that have historically experienced threat may have traits that are better matched to dealing with a collective threat such as COVID-19, whereas looser societies would experience more of a cultural mismatch, as evidenced in[12]. Another interpretation is that different threats may tighten different norms, namely those most relevant to overcoming the specific immediate threats: pandemics may make hygiene norms stronger while earthquakes may, instead, increase norms of helping. This would be consistent with an experimental study which found that a risk of collective loss increased the strength of norms concerning cooperation[13]. Over time, this would create a mosaic of norms that together correspond to the emergent notion of tightness. If correct, cultures that face a variety of threats will be those that end up the tightest. Another possibility is that pathogen threats, which are abstract and invisible, have particular characteristics and produce different tightening dynamics than threats which are concrete and visible (e.g., earthquakes, terrorism, or warfare)[31,32]. The former are harder to assess, potentially causing uncertainty and panic that may have led to egoistic behavior during early stages of the pandemic. Indeed, as extensively reported by the mass media, there was hoarding of resources in the early stages of the pandemic[33,34] and recent work finds evidence for the erosion of social trust[35].

These conclusions should also be considered in light of the limitations to our study. First, we use convenience samples (albeit both students and non-students). While this is unlikely to have substantial implications on our between-wave estimates, since the samples are broadly similar between the waves, it should be kept in mind when generalizing our findings to the broader populations. Specifically, it is possible that social norm change, or a lack thereof, occurred differently outside of cities, varied with socio-economic factors, or that younger people, who are overrepresented in our samples, experience fewer health-risks and our findings may not generalize to more senior people or those facing health issues. Second, our design allows us to avoid key endogeneity issues that are present in prior work, but cannot cleanly identify causal effects. More specifically, our first-stage analyses, comparing Wave 1 to Wave 2 averages, allows us to exclude reverse causality and country-constant confounders but it cannot exclude time-trends (e.g. changes in norm strength occurring over time irrespective of the pandemic). Our second-stage analyses, using perceived prevalence, fear, and government stringency to predict changes in the outcomes, reduces the possibility that such time-trends (or other confounding factors) are responsible for the observed changes as these would need to be correlated with our predictors and changes in social norms. In addition, we find little evidence for pre-existing time trends in tightness-looseness (see Methods and Fig. S7). Still, we do not have the power in the mechanisms analyses to detect small effects and cannot entirely identify causality.

## Methods

Our sample includes data from a first study wave collected before the breakout of the pandemic (April–December 2019, Wave 1[5]) and data from a second wave (March–July 2020, Wave 2) that we collected during the initial stages of the COVID-19 emergence. For comparability of samples across waves and among countries, we set out to collect data from approximately 200 college students at least in a major city in each country, which was achieved in all countries (Table S1). To assess the robustness of the country-level measures obtained from these samples, we complemented the main sampling strategy by collecting additional data from non-student samples.

When administering Wave 2, we aimed to collect data also from a subset of participants who took part in Wave 1 study. These participants were marked as "experienced" participants and were re-contacted (e.g. through laboratory recruitment systems). For six locations (Bosnia-Herzegovina, Canada, Colombia, Czech Republic, Italy, United States), we were able to recruit participants who had participated in Wave 1 but without matching their responses across waves. For two locations (Israel and Poland), we were able to uniquely identify participants and match their responses. Privacy and anonymity were nevertheless preserved in these samples. This allowed us to check whether experience of participation affects our findings. When specifically checking among participants matched across waves we find non-significant results that go in the same direction (see end of Methods).

In our analyses, we considered a response valid if a participant correctly passed an attention check placed at the end of the survey (i.e., participants had to click a specific item response). We discarded observations because of missing responses (4074 in Wave 1, 4660 in Wave 2) or failed attention checks (197 in Wave 1, 202 in Wave 2). We additionally excluded participants who declared an age under 18 (157 in Wave 1, 222 in Wave 2). The final dataset includes responses from 43 countries, 55 locations (six of which were sampled only in Wave 1, while only one sampled exclusively in Wave 2), and 30,431 valid respondents (see Table S1).

We used the survey administered in[5] to preserve comparability, with the sole addition of a small number of questions (at the end of the survey precluding any effects on the prior questions) regarding COVID-19 fear and prevalence, desired Tightness-Looseness measures, generalized trust, and risk aversion. The survey was translated into 30 different languages, following the standard practice of independent translation and back-translation. The study was conducted anonymously online using Qualtrics. The English version of the survey is publicly available as part of our pre-registration (https://osf.io/9ve4t). Our study is a survey therefore no randomization occurred and some of the investigators were not blinded to the study's hypotheses.

All participants gave their informed consent and we complied with all relevant ethical regulations. Approval of the study protocol was obtained from ethics committees and institutional review boards where required including for the University of Melbourne (Australia), Queen's University at Kingston (Canada), Universidad de los Andes (Colombia), Institute of Psychology, Czech Academy of Sciences (Czech Republic), Universidad San Francisco de Quito (Ecuador), United Research Ethics Committee of Psychology (Hungary), Monk Prayogshala (India), Trinity College Dublin (Ireland), Open University of Israel (Israel), LUISS University (Italy), United States International University - Africa (Kenya), Sunway University (Malaysia), University of Amsterdam (Netherlands), SWPS University (Poland), Universidade de Lisboa (Portugal), National University of Singapore (Singapore), University of Colombo (Sri Lanka), Koc University (Turkey), American University of Sharjah (United Arab Emirates), Brunel University London (United Kingdom), University of Kent (United Kingdom), University of South Carolina (United States of America), and New York University (United States of America). Ethical approval was not sought in countries where the approval received for the study conducted in Wave 1[5] was considered sufficient or where local legislation did not require ethical approval in the first place.

## Study preregistration
We pre-registered our study in two phases. Our initial pre-registration was submitted before data gathering (https://osf.io/zvdkt/) (March 23rd 2020) and contained a design and provisional data analysis plan. Due to the short timeframe before data collection began, the analysis plan was only provisional. Our second pre-registration, which was submitted after the data were collected but before the data were examined or analyzed (October 22nd 2020), contains a detailed analysis plan that we completely followed (https://osf.io/9ve4t).

The hypotheses that we pre-registered and test are the following:
- H1: Tightness-Looseness levels in Wave 2 will be higher on average than in Wave 1.
- H2a: Perceived threat will be positively associated with change in tightness.
- H2b: Perceived prevalence will be positively associated with change in tightness.
- H2c: A stricter governmental response will be positively associated with change in tightness.
- H3a: Punishments, on average, are perceived as more appropriate.
- H3b: People are likelier to engage in punishing norm violations.

In addition to the aforementioned hypotheses, we investigate the differences in situation specific norms and a set of items measuring hand hygiene norms between waves 1 and 2 to provide a fuller understanding in social norm changes. Furthermore, to study the mechanisms for hand hygiene norms and punishment change, we complement our analyses by exploring the moderating role of perceived threat, COVID-19 prevalence, and governmental stringency on the change in hand hygiene norms and frequency of punishment, both of which show consistent changes from Wave 1 to Wave 2.

## Survey measures
We measured the following variables through survey questions. These were elicited in both Wave 1 and Wave 2 unless stated otherwise.

**Tightness-looseness scores.** We compute tightness-looseness scores (TL) following individual-level standardization as in past work[3,5]. Standardization is needed to adjust for cross-cultural variation in response sets given that some cultures are more likely to provide extreme responses or acquiesce to survey items than others[3,36]. Following guidelines from cross-cultural psychology[36,37], and from data published in the first wave[5], we calculate appropriateness scores by averaging each individual's responses to a large set of heterogeneous items (i.e. 50 appropriateness items that all used the same response scale, from extremely inappropriate to extremely appropriate). This score is then subtracted from participants' responses in the tightness-looseness questionnaire (6 items from ref. 3). The final individual TL scores are computed by averaging the adjusted 6 items. After transformation, TL scores display an overall average $\bar{x} = 1.85$, standard deviation $s = 0.81$, min $= -2.26$, max $= 5.25$. Differently from[5], we did not impute missing TL data. This resulted in tiny differences in TL scores between studies (difference between mean TL scores $= 0.01$) that do not affect the validity of our results. The correlation between our TL scores and those appearing in[5] is essentially perfect (Spearman test, $r = 0.997$, $p < 0.001$). Standardizing tightness-looseness scores does not affect our results (checked for all tightness-looseness analyses reported in the manuscript). Furthermore, the correlation between standardized and non-standardized measures of TL is high and significant ($r = 0.84$ for Wave 1 measures, $r = 0.85$ for Wave 2 measures, $p < 0.001$ in both cases).

Given our empirical interest in assessing the change in tightness-looseness associated with the emergence of the pandemic, we also checked whether TL scores changed or not between 2000–2003 (Wave 0), using data from[3], and 2019 (Wave 1)[5], and 2020 (Wave 2). We find that tightness-looseness scores have remained unchanged in almost all countries since 2000–2003 (Wave 0 to Wave 1: $r = 0.89$; Wave 0 to Wave 2: $r = 0.88$, all $p < 0.001$) and that there is strong stability in the ordering of countries (Kendall rank test, $t = 0.752$, $p < 0.001$, Fig. S7 panels A, B) implying that TL is a stable measure. More formally, to check whether trends in TL scores were similar across our countries pre-pandemic, with respect to their

post-pandemic COVID-19 intensity, we use the following model:

$$TL_{ct} = \alpha + \beta Covid\ Severity + \lambda_1 Wave_1 + \lambda_2 Wave_2 + \delta_1 Covid\ Severity * Wave_1 + \delta_2 Covid\ Severity * Wave_2 \quad (1)$$

Where $TL$ indicates tightness-looseness from country $c$, at time $t$; $Wave$ are dummy variables indicating the study wave (Wave 1 or Wave 2; Wave 0 is the baseline), and $Covid\ Severity$ is fear of COVID-19, perceived cases, actual COVID-19 cases, or COVID-19 deaths (we check each sequentially). If there are no systematic differences in trend pre-pandemic then $\delta_1 = 0$. This would indicate that countries that were later affected by the pandemic with heterogenous intensities had TL change that followed the same pattern between Wave 0 and Wave 1. We find no evidence for systematic differences in trends of TL scores between 2000–2003 and 2019 according to later COVID-19 severity (Table S14).

**Situation-specific norms.** Participants' appropriateness ratings are measured with their responses to five scenarios that cover potential norm-violating behavior in several domains concerning cooperation and out-of-place everyday behavior (see Analysis Plan of the pre-registration Analysis Plan). Ratings of the appropriateness of each item were elicited through a six-point scale, ranging from extremely inappropriate (coded 0) to extremely appropriate (coded 5). Average rating across countries is $\bar{x} = 1.13$, standard deviation $s = 0.60$, min = 0, max = 5.

**Metanorm scenarios.** Metanorms were collected for each situation (five in total) based on survey items reported in our pre-registered analyses plan. Items covered five different punishment behaviors for each situation (hence, a total of 25 items, see Analysis Plan of pre-registration Analysis Plan), which are: verbal and physical confrontation, gossip, non-action (reverse coded) and ostracism, and we collected participants' ratings of the appropriateness of each. Appropriateness was elicited through a six-point scale, ranging from extremely inappropriate (coded 0) to extremely appropriate (coded 5). Each punishment behavior is used as a separate dependent variable. Average appropriateness across countries is $\bar{x} = 2.22$, standard deviation $s = 1.25$, min = 0, max = 5.

**Punishing.** We consider three survey items eliciting the frequency at which respondents engaging in confronting, gossiping, and ostracizing someone who behaves inappropriately. Frequency of punishment was elicited using a five-point scale ranging from never (coded 1) to always (coded 5). We analyzed these all together (with mixed effects at the scenario level) and also conducted separate analyses for each item separately. Average frequency of punishment across countries is $\bar{x} = 2.98$, standard deviation $s = 0.59$, min = 1, max = 5.

**Hand washing norms.** Our survey asked participants in which of six situations they think people should wash hands. These situations are: before eating a meal, after eating a meal, after defecating, after urinating, when they come home, and after shaking someone's hand. Hand washing norms are analyzed using as both the number of situations considered as appropriate (number of ticks) as well as whether a participant considered a given situation as appropriate (participant ticked or not a given situation). Because of a translation mistake in our survey, one country (Indonesia) has been excluded from all the analyses of these items. Average number of appropriate situations across countries was $\bar{x} = 4.28$, standard deviation $s = 1.30$, min = 0, max = 6.

**Fear of COVID-19.** Our measure of COVID-19 fear comes from the Wave 2 survey. In particular, respondents answered three items: "How concerned are you by the spread of the new Coronavirus (COVID-19)?" "How much fear do you have by the spread of the Coronavirus?" "How

dangerous do you think the Coronavirus is?". Participants responded on a six-point scale. We then compute the average over items. Average COVID-19 fear is $\bar{x} = 4.42$, standard deviation $s = 0.41$, min = 3.42, max = 5.20. Following our pre-registration, we checked internal consistency of the items listed above reporting (Cronbach's $\alpha = 0.84$). We additionally computed Cronbach's alphas for each country separately. Estimated values range from 0.58 (Kenya) to 0.90 (Poland) (see below for full list). The cross-country average is 0.80 ($s = 0.07$) which is close to the value obtained when merging all countries in our sample. Since estimated Cronbach alphas fall within the range of satisfactory internal consistency, throughout our main analyses, we averaged these items to create a single variable at the individual level. The only country with alpha <0.60 is Kenya; all our analyses reported in the manuscript are robust and do not substantially change when excluding Kenya from the dataset.

The full list of countries' alphas is: ARE: 0.81, ARG: 0.76, ARM: 0.82, AUS: 0.78, BIH: 0.83, BRA: 0.79, CAN: 0.82, CHL: 0.80, CHN: 0.77, COL: 0.80, CZE: 0.85, DEU: 0.86, ECU: 0.75, ESP: 0.79, EST: 0.87, FIN: 0.84, GBR: 0.86, GRC: 0.85, HUN: 0.87, IDN: 0.83, IND: 0.71, IRL: 0.84, ISL: 0.77, ISR: 0.90, ITA: 0.86, JPN: 0.85, KEN: 0.58, KOR: 0.87, LKA: 0.63, MYS: 0.66, NGA: 0.65, NLD: 0.78, POL: 0.91, PRT: 0.88, RUS: 0.77, SAU: 0.84, SGP: 0.82, SWE: 0.80, TUR: 0.84, UKR: 0.89, USA: 0.82, VNM: 0.83. PER: items missing due to error in data collection.

**Perceived COVID-19 prevalence.** Our measure of disease prevalence was elicited with the Wave 2 survey question "What percent of people living in your province do you think have been infected with COVID-19? Please do not look up actual statistics to answer this question—just enter your best guess" (0–100). Average perceived COVID-19 prevalence across countries is $\bar{x} = 21.87$, standard deviation $s = 7.05$, min = 8.53, max = 42.65.

### External measures
We measured the following variables through external data sources that we matched with our survey data.

**Stringency Index.** Our measure of the intensity of government response to COVID-19 is the Stringency Index from the Oxford COVID-19 Government Response Tracker[22]. The measure contains indicators reporting the severity of containment and closures (e.g. school and workplace closures and restrictions on gathering size; see items C1-C8 in ref. [23]) and public information campaigns (item H1 in ref. [23]). The Stringency Index can vary between 0 and 100. We match participants' responses to our survey with Stringency Index data calculated on the same day. Average stringency across countries is $\bar{x} = 78.12$, standard deviation $s = 13.54$, min = 32.77, max = 99.48.

**Deaths and cases.** We use COVID-19 deaths and cases per million from Our World in Data[38] (downloaded November 2020). Data were matched with participants' responses to our survey based on day of response (thus case and deaths data run from March–July 2020). Average of deaths across countries and periods is $\bar{x} = 47.88$ per million, standard deviation $s = 103.70$, min = 0.05, max = 481.99. Average of cases across countries and periods is $\bar{x} = 834.95$ per million, standard deviation $s = 1067.72$, min = 1.98, max = 4389.68.

### Computed measures
The following measures were computed based on changes between Wave 1 and Wave 2. In addition to the pre-registered test ΔTightness-Looseness, we did this only for those variables that showed robust changes between the waves (see Analyses).

**ΔTightness-looseness, Δpunishing, and Δhand washing.** When computing change in TL, we averaged individual scores for each

country and compute the difference between Wave 2 and Wave 1 values (Wave 2–Wave 1). A similar procedure is followed for computing change in other items. For hand washing and punishing items (frequency of punishment) we computed changes across waves both for each individual item and for the average of all items.

## Analyses

We started by analyzing the between-wave changes in Tightness-Looseness, situation-specific norms, metanorms, punishing, and hand washing norms. Then, for those changes that are shown to be robust (across sub-items and model specifications, including with random slopes and with controls for COVID-19 cases and deaths), we examine the mechanisms predicting a change in our variables of interest (ΔTightness-Looseness, Δpunishing, and Δhand washing). The models used for both stages are outlined below. In addition to these models, we replicated all of our analyses with the addition of random slopes to allow for country-level variation of the effect associated with COVID-19 pandemic. For these, we additionally report $\tau_{11}$, the variance of the main parameter of interest (*Wave* 2) to shed light on the heterogeneity of the effect due to COVID-19 pandemic among countries. Moreover, we also conducted these analyses controlling for deaths and cases (adjusted to each country population size) to account for the different levels of COVID-19 pandemic across the countries and this does not affect our results. For all coefficient estimates we report the results from two-sided *t*-tests. All tests meet the relevant assumptions. We do not adjust for multiple comparisons.

**Tightness-looseness, situation-specific norms, and punishing.** We use multilevel models with random intercepts at the individual ($n \approx 29{,}000$), city ($n = 55$), and country ($n = 43$) level. Put formally, to test Hypothesis 1, we estimate the following multilevel model with varying intercepts at the country ($c$), city ($k$) and individual ($i$) level:

$$TL_{cki} = \beta_0 + \beta_{0c} + \beta_{0k} + \beta_1 Wave\,2_{cki} + \delta \mathbf{Z}_{cki}, \qquad (2)$$

where $\mathbf{Z}$ is the vector of control variables to account for possible between-wave sample variation (age, gender, and student/non-student status), Wave 2 is a dummy variable taking value 1 when an observation was collected in Wave 2 and 0 otherwise. Our analyses for situation-specific norms, punishing, and hand washing norms follow the same model structure with the dependent variable changed to those variables.

**Metanorms.** We use multilevel models with random intercepts at the country ($c$), city ($k$), scenario ($s$), and individual ($i$) levels and implement the following model specification:

$$A_{cksi} = \beta_0 + \beta_{0c} + \beta_{0k} + \beta_{0s} + \beta_1 Wave\,2_{cksi} + \beta_2 N_{cks} + \delta \mathbf{Z}_{cksi} \qquad (3)$$

where $A$ is the appropriateness score given by individual $i$ to the punishment scenario $s$, in country $c$, city $k$. $N$ is the average appropriateness at the location level that participants have given to the norm violation of scenario $s$ (see also Methods in ref. 3) and $\mathbf{Z}$ is a vector of demographic controls (age, gender, and student/non-student status).

**Hand washing norms.** We used two approaches to test hand washing norms. First, to model the number of ticked categories we use the same model structure as Eq. 2 but with the dependent variable replaced with the number of ticks given by participant $i$, in county $c$, and city $k$. Second, to test the probability of ticking each single situation we use a multilevel logit regression with random intercepts at the country and city level:

$$\log(H_{cki}) = \beta_0 + \beta_{0c} + \beta_{0k} + \beta_1 Wave\,2_{cki} + \delta \mathbf{Z}_{cki}, \qquad (4)$$

Where $H$ is the odds of participant $i$, in country $c$, and city $k$, ticking that it is appropriate to wash hands for a given setting. $\mathbf{Z}$ is a vector of demographic controls (age, gender, and student/non-student status).

**ΔTightness-looseness, Δpunishing frequency, and Δhand washing.** These analyses are conducted using heteroskedasticity-robust OLS regressions with observations at the country level. Observations are country-level as the dependent variable is Wave 1 to Wave 2 change in a given country. We do not use city-level because in a small number of countries different cities were sampled between Wave 1 and Wave 2. Put formally we estimate the following model for ΔTightness-Looseness:

$$\Delta TL_c = \beta_0 + \beta_1 PC_c + \beta_2 Fear_c + \beta_3 SI_c + \delta \mathbf{Z}_c \qquad (5)$$

where *Fear* is fear of COVID-19, *PC* is perceived cases of COVID-19, and *SI* is the Stringency Index from the Oxford COVID-19 Government Response Tracker.

We performed similar analyses for the change in hand washing and punishment. In particular, for the former, we conducted analyses for the change in the number of ticks for (i) all items, (ii) specifically for items that were not directly related to the COVID-19 pandemic (before meal, after meal, after defecating, and after urinating), (iii) specifically for items that are directly related to the pandemic (after shaking hands and after coming home), and (iv) each item separately that is directly related to the pandemic (Table S12).

For the items measuring punishing frequency, we estimate the change in responses for each single item individually (Table S9), and change in the mean of all of our 3 items (grand mean change) (Table S10).

## Tightness-looseness change for tracked participants

We were able to perfectly match responses to our survey across waves for two locations in our sample: Israel and Poland. Below, we report the results from a robustness check aimed to test tightness score decrease.

For our Israel sub-sample of tracked participants ($N = 57$), tightness scores decrease on average of 0.16 (Cohen's $d = 0.17$, Wilcoxon paired samples $r = 0.172$), yet the change is not significant (Wilcoxon paired samples test, $V = 30$, $p = 0.195$). For our Poland sub-sample ($N = 10$), tightness scores decrease by about 0.12 (Cohen's $d = 0.15$), but the change is not significant (Wilcoxon paired samples test, $V = 30$, $p = 0.85$). We interpret results from our sub-samples as highly noisy but consistent with our general results from the full dataset showing a small decrease in tightness scores.

For 6 locations (Bosnia-Herzegovina, Canada, Colombia, Czech Republic, Italy, United States), we were able to distinguish responses coming from participants who previously participated in the first wave, but were not able to match the id of each responses. By running multilevel linear regression models, we report evidence of no significant change in tightness-looseness scores for these sub-populations ($b = 0.046$, $p = 0.222$).

## Power analysis

The main aim of this study was to examine whether the pandemic was associated with a systematic change in tightness-looseness (TL) scores compared to pre-pandemic scores. To make sure that our sample is large enough to detect small changes in TL, we compute the power achieved based on the mixed effects model in Eq. 2. We adopt the common convention that a small effect be equivalent to a Cohen's $d$ of at least 0.10. From our sample, it means that the average TL score changes by at least 10% of its standard deviation, that is a change in TL of 0.08 (TL $s = 0.80$). By using the R package "simr", we estimate the 95% CI of achieved power from the model in Eq. 2 to be 95% CI = [96.38; 100] (predictor "Wave2", $\alpha = 0.05$, 100 simulations).

We then perform sensitivity analysis to provide evidence of sufficient achieved power for models testing the change in TL scores. Given a sample of 28,374 individuals, a significance level of α = 0.05, and a desired power 0.80, we estimate the minimum detectable change in raw TL scores of 0.025 (equivalent to Cohen's $d$ = 0.03).

We also perform sensitivity analysis for the proposed mechanisms variables (Eq. 5). Given a sample of 41 countries, a significance level of α = 0.05, and a desired power 0.80, we estimate the minimum detectable effect size f². Results show that the minimum effects that could be detected are of medium to large size f² = 0.2 (two sided) for the proposed mediating variables.

### Equivalence tests

We performed equivalence tests for all the Wave 1 to Wave 2 change analyses following the two one-sided test (TOST) procedure[23-25]. To set the smallest effect size of interest (SESOI) it is recommended to use substantive motivations (e.g. prior findings in the literature)[23,24]. Yet, for our analyses, we were unable to identify clear substantive bases for setting the SESOI. For instance, comparable meta-norm measures do not exist, to our knowledge, while for tightness-looseness, there is only one other source for comparable large-scale cross-country data[3] but this is solely available in a transformed form making a comparison in mean change to our waves meaningless (see Supplementary Note 1). Given this absence of comparable prior empirical evidence for setting the SESOIs, we consider a Cohen's $d$ = 0.10 as the SESOI for changes in our measures over time. While for all mechanism analyses, we considered standardized betas as effect size measure, and consider a threshold of $\beta$ = ±0.10 (a small effect size[26]) as the SESOI benchmark for all mechanisms tested.

We conducted the TOST procedure (set at the 5% significance level) using the coefficients and standard errors derived from the model estimates displayed in the main text and supplementary materials. For example, when analyzing the SESOI for TL, we estimate the equivalent change Δ in the raw scale corresponding to $d$ = 0.10. The coefficient estimate and standard error are drawn from Model 1 (Table S2) and the TOST procedure is applied. The SESOIs of all other norm measures are calculated by applying the same reasoning and the TOSTs are conducted in the same way. For each equivalence test, we report the smallest magnitude $t$-value from among the two one-sided tests performed.

**Tightness-looseness.** We find a significant difference between our estimate of TL change and the SESOI (Δ = ±0.08, $t$(28369) = 5.53, $p$ < 0.001) such that the relevant coefficient ($b$ = −0.028, 90% CI = [−0.047; −0.009]) is contained within the upper and lower equivalence bounds. This indicates that although there is a significant decrease in TL from Wave 1 to Wave 2 the change is statistically equivalent.

**Situation-specific norms.** We find a significant difference between our estimate of situation-specific norms change and the within-country SESOI (Δ = ±0.06, $t$(142531) = 7.802, $p$ < 0.001) such that the relevant coefficient ($b$ = −0.017, 90% CI = [−0.028; −0.006]) is contained within the upper and lower equivalence bounds. This indicates that while we find a significant decrease in situation-specific norms from Wave 1 to Wave 2, the change is statistically equivalent.

**Metanorms.** We find a significant difference between our estimate of metanorms change and the SESOI (Δ = ±0.05, $t$(484665) = −12.925, $p$ < 0.001) such that the relevant coefficient ($b$ = 0.006, 90% CI = [−0.001; 0.012]) is contained within the upper and lower equivalence bounds. This implies that the change in metanorms is not significant from Wave 1 to Wave 2 and statistically equivalent.

**Punishing frequency.** We find a significant difference between our estimate of punishing frequency change and the SESOI (Δ = ±0.1, $t$(85490) = 9.603, $p$ < 0.001) such that the relevant coefficient ($b$ = −0.034, 90% CI = [−0.047; −0.022]) is contained within the upper and lower equivalence bounds. This means that, although we find a statistically significant decrease in punishing frequency, the change is statistically equivalent.

**Hand washing norms.** We find a significant difference between our estimate of hand washing norms change and the SESOI (Δ = ±0.13, $t$(28134) = −49.84, $p$ < 0.001) such that the relevant coefficient ($b$ = 0.420, 90% CI = [0.390; 0.450]) is above the upper equivalence bound. This implies that the change in hand washing norms is significant from Wave 1 to Wave 2 and not statistically equivalent.

**Mechanism analyses.** When running the equivalence tests for the factors included in the mechanism analysis of the change in *TL scores*, we find that all standardized coefficients of our factors (Fear of COVID-19, $\beta$ = −0.283, 90% CI [−0.503; −0.063]; Perceived Prevalence, $\beta$ = −0.201, 90% CI [−0.526; 0.124]; Gov. Stringency, $\beta$ = −0.036, 90% CI [−0.205; 0.133]) overlap with either the upper or lower equivalence bounds. This means that there is insufficient evidence to conclude a negligible effect.

The same analyses run for the change in hand washing norms give similar results in terms of equivalence. The coefficient associated with Fear of COVID-19 ($\beta$ = 0.352, 90% CI = [0.087; 0.618]), Perceived Prevalence ($\beta$ = 0.343, 90% CI [0.017; 0.669]) as well as Gov. Stringency ($\beta$ = −0.058, 90% CI [−0.333; 0.216]) overlap with either the upper or lower bound of the equivalence interval indicating that there is insufficient evidence to conclude a negligible effect.

Likewise, results from the equivalence tests for the change in punishing frequency show that the coefficient associated with Fear of COVID-19 ($\beta$ = −0.080, 90% CI = [−0.2314; 0.071]), Perceived Prevalence ($\beta$ = 0.008, 90% CI [−0.241; 0.257]) as well as Gov. Stringency ($\beta$ = −0.096, 90% CI [−0.255; 0.062]) overlap with either the upper or lower bound of the equivalence interval indicating that there is insufficient evidence to conclude a negligible effect.

### Reporting summary

Further information on research design is available in the Nature Portfolio Reporting Summary linked to this article.

## Data availability

The data generated in this study have been deposited in the Open Science Framework (https://doi.org/10.17605/OSF.IO/STKFR). Non-experimental data included in our datasets (i.e., intensity of government response to COVID-19 is the Stringency Index, COVID-19 deaths and cases per million) are taken from the Oxford COVID−19 Government Response Tracker[22] and Our World in Data[38] (downloaded November 2020). Wave 0 data are from[3] and Wave 1 data are from[5].

## Code availability

The survey and analysis code are available at the Open Science Framework (https://doi.org/10.17605/OSF.IO/STKFR).

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

## Acknowledgements

Knut and Wallenberg Grant "How do human norms form and change?" 2016.0167. (G.An.). The Swedish Research Council grant "Norms & Risk: Do social norms help dealing with collective threats" 2021-06271 (G.An.). Ministero dell'Istruzione dell'Università e della Ricerca, PRIN 2017, prot. 20178TRM3F (D.B.). Universidad de Los Andes, Fondo Vicerrectoría de Investigaciones (J.-C.C.). Ministry of Innovation and Technology of Hungary, National Research, Development and Innovation Fund NKFIH-OTKA K135963 (M.F.). Grant 23-061770 S of the Czech Science Foundation (M.H. and S.G.). RVO: 68081740 of the Institute of Psychology, Czech Academy of Sciences (M.H. and S.G.). RA Science Committee, research project N.20TTSH-070 (A.Gr. and N.Khac.). Open University of Israel, 511687 (R.N.). HSE University Basic Research Program (E.O.). Project BASIC (PID2022-141802NB-I00) funded by MCIN/AEI/10.13039/501100011033 and by "ERDF A way of making Europe" (A.Sá.). US Army Research Office Grant W911NF-19-1-910281 (B.S.). Netherlands Organisation for Scientific Research, 019.183SG.001 (E.S.). Netherlands Organisation for Scientific Research, VI.Veni.201 G.013 (E.S.). European Commission, Horizon 2020-ID 870827 (E.S.). UKRI Grant "Secret Power" No. EP/X02170X/1 awarded under the European Commission's "European Research Council - STG" Scheme (G.A.T.).

## Author contributions

G.An., A.Sz. and A.Gu. designed the study and wrote the manuscript. A.Gu. analyzed the data. K.E., M.G., A.Gl. and M.T.L. provided critical input on the study and/or the manuscript. All other authors arranged translations where required, gave feedback on wording of items, collected data, and reviewed the manuscript.

## Competing interests

The authors declare no competing interests.

## Additional information

Giulia Andrighetto[1,2,3,79] ✉, Aron Szekely[1,4,79], Andrea Guido[1,2,5], Michele Gelfand[6], Jered Abernathy[7], Gizem Arikan[8], Zeynep Aycan[9,10], Shweta Bankar[11], Davide Barrera[4,12], Dana Basnight-Brown[13], Anabel Belaus[14,15], Elizaveta Berezina[16], Sheyla Blumen[17], Paweł Boski[18], Huyen Thi Thu Bui[19], Juan Camilo Cárdenas[20,21], Đorđe Čekrlija[22,23], Mícheál de Barra[24], Piyanjali de Zoysa[25], Angela Dorrough[26], Jan B. Engelmann[27], Hyun Euh[28], Susann Fiedler[29], Olivia Foster-Gimbel[30], Gonçalo Freitas[31], Marta Fülöp[32,33], Ragna B. Gardarsdottir[34], Colin Mathew Hugues D. Gill[16,35], Andreas Glöckner[26], Sylvie Graf[36], Ani Grigoryan[37], Katarzyna Growiec[18], Hirofumi Hashimoto[38], Tim Hopthrow[39], Martina Hřebíčková[36], Hirotaka Imada[40], Yoshio Kamijo[41], Hansika Kapoor[42], Yoshihisa Kashima[43], Narine Khachatryan[37], Natalia Kharchenko[44], Diana León[45], Lisa M. Leslie[30], Yang Li[46], Kadi Liik[47], Marco Tullio Liuzza[48], Angela T. Maitner[49], Pavan Mamidi[11], Michele McArdle[8], Imed Medhioub[50], Maria Luisa Mendes Teixeira[51], Sari Mentser[52], Francisco Morales[53], Jayanth Narayanan[54], Kohei Nitta[55], Ravit Nussinson[56,57], Nneoma G. Onyedire[58], Ike E. Onyishi[58], Evgeny Osin[59], Seniha Özden[9], Penny Panagiotopoulou[60], Oleksandr Pereverziev[61], Lorena R. Perez-Floriano[62], Anna-Maija Pirttilä-Backman[63], Marianna Pogosyan[64], Jana Raver[65], Cecilia Reyna[14], Ricardo Borges Rodrigues[66], Sara Romanò[12], Pedro P. Romero[67,68], Inari Sakki[63], Angel Sánchez[69,70], Sara Sherbaji[49,71], Brent Simpson[7], Lorenzo Spadoni[72], Eftychia Stamkou[73], Giovanni A. Travaglino[40], Paul A. M. Van Lange[74], Fiona Fira Winata[75], Rizqy Amelia Zein[75], Qing-peng Zhang[76] & Kimmo Eriksson[2,77,78]

[1]Institute of Cognitive Sciences and Technologies, National Research Council of Italy, Rome, Italy. [2]Institute for Futures Studies, Stockholm, Sweden. [3]Institute for Analytical Sociology, Linköping University, Linköping, Sweden. [4]Collegio Carlo Alberto, Turin, Italy. [5]CEREN EA 7477, Burgundy School of Business, Université Bourgogne Franche-Comté, Dijon, France. [6]Graduate School of Business and Department of Psychology, Stanford University, Stanford, USA. [7]Department of Sociology, University of South Carolina, Columbia, USA. [8]Department of Political Science, Trinity College Dublin, Dublin, Ireland. [9]Department of Psychology, Koç University, Istanbul, Turkey. [10]Faculty of Management, Koç University, Istanbul, Turkey. [11]Ashoka University, Sonipat, India. [12]Department of Culture, Politics, and Society, University of Turin, Turin, Italy. [13]United States International University – Africa, Nairobi, Kenya. [14]Instituto de Investigaciones Psicológicas (IIPsi), Consejo Nacional de Investigaciones Científicas y Técnicas (CONICET); CABA, Córdoba, Argentina. [15]Facultad de Psicología, Universidad Nacional de Córdoba (UNC), Córdoba, Argentina. [16]Sunway University, Bandar Sunway, Malaysia. [17]Departamento de Psicología, Pontificia Universidad Católica del Perú, Lima, Perú. [18]SWPS University, Warsaw, Poland. [19]Hanoi National University of Education, Hanoi, Vietnam. [20]Universidad de los Andes, Bogota, Colombia. [21]University of Massachusetts Amherst, Amherst, USA. [22]Faculty of Philosophy, University of Banja Luka, Banja Luka, Bosnia and Herzegovina. [23]Institute of Psychology, University of Greifswald, Greifswald, Germany. [24]Centre for Culture and Evolution, Brunel University London, Uxbridge, UK. [25]Faculty of Medicine, University of Colombo, Colombo, Sri Lanka. [26]Department of Psychology, University of Cologne, Cologne, Germany. [27]Center for Research in Experimental Economics and Political Decision Making (CREED), Amsterdam School of Economics, University of Amsterdam, Amsterdam, The Netherlands. [28]Gies College of Business, University of Illinois at Urbana-Champaign, Champaign, USA. [29]Vienna University of Economics and Business, Vienna, Austria. [30]Stern School of Business, New York University, New York, USA. [31]Instituto de Ciências Sociais, Universidade de Lisboa, Lisboa, Portugal. [32]HUN-REN Institute of Cognitive Neuroscience and Psychology, Research Centre of Natural Sciences, Budapest, Hungary. [33]Institute of Psychology, Karoli Gáspár University of the Reformed Churches, Budapest, Hungary. [34]Faculty of Psychology, University of Iceland, Reykjavik, Iceland. [35]Universal College Bangladesh, Dhaka, Bangladesh. [36]Institute of Psychology, Czech Academy of Sciences, Brno, Czech Republic. [37]Department of Personality Psychology, Yerevan State University, Yerevan, Armenia. [38]Osaka Metropolitan University, Osaka, Japan. [39]School of Psychology, University of Kent, Canterbury, UK. [40]Royal Holloway, University of London, Egham, UK. [41]Waseda University, Tokyo, Japan. [42]Department of Psychology, Monk Prayogshala, Mumbai, India. [43]Melbourne School of Psychological Sciences, The University of Melbourne, Melbourne, Australia. [44]Kyiv International Institute of Sociology, Kyiv, Ukraine.

[45]DeJusticia, Bogotá, Colombia. [46]Nagoya University, Nagoya, Japan. [47]School of Natural Sciences and Health, Tallinn University, Tallinn, Estonia. [48]Department of Medical and Surgical Sciences, "Magna Græcia" University of Catanzaro, Catanzaro, Italy. [49]Department of Psychology, American University of Sharjah, Sharjah, United Arab Emirates. [50]Department of Finance and Investment, Imam Mohammad Ibn Saud Islamic University, Riyadh, Saudi Arabia. [51]Presbyterian Mackenzie University, São Paulo, Brazil. [52]The Hebrew University of Jerusalem, Jerusalem, Israel. [53]Universidad de los Andes, Santiago, Chile. [54]Northeastern University, Boston, USA. [55]Ritsumeikan University, Shiga, Japan. [56]Department of Education and Psychology, The Open University of Israel, Ra'anana, Israel. [57]Institute of Information Processing and Decision Making (IIPDM), University of Haifa, Haifa, Israel. [58]Department of Psychology, University of Nigeria, Nsukka, Nigeria. [59]HSE University, Moscow, Russia. [60]Department of Education and Social Work, University of Patras, Patras, Greece. [61]POLLSTER, Kiev, Ukraine. [62]Universidad Diego Portales, Santiago, Chile. [63]Faculty of Social Sciences, Social Psychology, University of Helsinki, Helsinki, Finland. [64]Leadership and Management, Amsterdam Business School (ABS), University of Amsterdam, Amsterdam, The Netherlands. [65]Queen's University at Kingston, Ontario, Canada. [66]Centro de Investigação e Intervenção Social, Instituto Universitário de Lisboa (ISCTE-IUL), Lisbon, Portugal. [67]School of Economics, Universidad San Francisco de Quito, Quito, Ecuador. [68]Experimental and Computational Economics Lab (ECEL), Universidad San Francisco de Quito, Quito, Ecuador. [69]Grupo Interdisciplinar de Sistemas Complejos (GISC), Departamento de Matemáticas, Universidad Carlos III de Madrid, Leganés, Spain. [70]Instituto de Biocomputación y Física de Sistemas Complejos (BIFI), Universidad de Zaragoza, Zaragoza, Spain. [71]Department of Anthropology, University College London, London, UK. [72]Department of Economics and Law, University of Cassino and Southern Lazio, Cassino (FR), Italy. [73]Department of Psychology, University of Amsterdam, Amsterdam, The Netherlands. [74]Department of Experimental and Applied Psychology, Vrije Universiteit, Amsterdam, The Netherlands. [75]Faculty of Psychology, Universitas Airlangga, Surabaya, Indonesia. [76]Guangzhou University, Guangzhou, P. R. China. [77]Center for Cultural Evolution, Stockholm University, Stockholm, Sweden. [78]Malardalens University, Vasteras, Sweden. [79]These authors contributed equally: Giulia Andrighetto, Aron Szekely. ✉e-mail: giulia.andrighetto@istc.cnr.it

