## [Peer Review File · Nature Communications]

Reviewers' Comments:

Reviewer #1:

Remarks to the Author:

This paper tests the predicts of cultural tightness-looseness theory by seeing how norms changed in the early months of the COVID pandemic. It takes advantage of an earlier published experiment on cross-cultural variation in norms, and sees how those norms (and enforcement thereof) differed between the original data collection (up to late 2019) versus April-June 2020 when many countries were first closing down due to COVID. They find that contrary to expectations, norms became *less* tight during the pandemic (except for handwashing for obvious reasons). This speaks against some of the predictions of cultural tightness-looseness theory. Overall, this was an interesting paper that was a pleasure to read. Plus, it provides an important test of an influential theory using a global natural experiment and a rich dataset. Overall I think the quality and broad interest make it a good candidate for publication in Nature Communications.

My biggest question is whether these data are best looked at in April to June 2020, and whether there would have been sufficient time for cultural change to occur given that the pandemic really started in March. Ideally they would have also collected data a little later too, such as late 2020 or early 2021, to allow for sufficient cultural change to have occurred and test three time points (pre-pandemic, early pandemic, later pandemic). However, I recognize that that is no longer a possibility, so the question is really whether cultural change would have occurred within the first few months of the pandemic. If there had been no change in norms (or enforcement thereof), the results would have been inconclusive. However, the results are significant in the opposite direction than predicted, which is at least evidence against a simple "any threat leads to general tightening" explanation. As such, I don't think it's a major problem that there are only data from the early pandemic. However, this does require more discussion to justify whether norm change could occur on such short timescales. There is some discussion, but the authors should do more to justify the timescale they looked at, and what it might have looked at had they collected the second wave a few months later to allow for more time for cultural change.

I have two further minor comments that are easily resolved.

First, in some places in the supplementary material it says they "disposed of" some variables. Does this mean that they excluded these variables? If so, this requires justification. Or is this just a funny way of saying that they use those variables, perhaps a translation error?

Second, in table S5 it should clarify what exactly the dependant variable is, e.g., the appropriateness of that action.

Signed,
Pat Barclay

Reviewer #2:

Remarks to the Author:

This paper tests whether, early in the COVID pandemic, the general tightness of societies increased and whether the strength of specific norms increased. As they report, in many cases there was no effect of the COVID pandemic. The two exceptions were that hand hygiene norms got stronger (which is relevant in general to transmissible diseases) and, surprisingly, the samples studied tended to get overall looser in their approach to norms.

I think this is a strong paper. It asks several related, clear questions. It has a huge sample, with people from throughout the world. It is pre-registered. I also appreciate that the authors of the study are working to publish findings that, while not contradicting their theory, are nonetheless perhaps a bit uneasy for their theory to account for.

The only major concern I had with the project was the standardization: The tightness-looseness items are standardized based on ratings from 50 "appropriate" questions. Why is this necessary? No explanation is given. Instead, the reader is referred to an earlier paper. But I didn't follow the conceptual rationale in that paper either; primarily, the standardization section of that paper also

just describes what was mechanically done. Since this is one of the main dependent variables, readers should fully understand why it was calculated as it was. Personally, I think it's a bit strange to directly subtract scales from each other when the anchors are different.

I will note that I do not have expertise in multilevel regression, so I can not assess the appropriateness of their modeling choices.

I support publication with revisions.

Minor typo: in the "meta norms and punishing" section in the results, you should refer to Fig 1C, not 1D.

Reviewer #3:

Remarks to the Author:

Summary of the paper:

=====

This paper uses two large global (43 countries) surveys to understand the impact of a global shock as big as COVID-19 on social norms and behaviors. The paper reports the results of surveying >30,000 people before (April-Dec 2019) and in the initial phase (April-June 2020) of COVID-19. The surveys contained various questions about the social norms. The paper uses tightness-looseness theory and the data from the two surveys to study the change of social norms. A difference-in-difference analysis is also done using the difference in the responses from the two surveys as the dependent variable. The results show that COVID-19 strengthened hand-washing norms but did not change most other norms.

Comments:

=====

The paper is well written and reports on a highly impressive large-scale data analysis on the societal impacts of COVID-19. Kudos to the entire team of authors on writing this paper! While the scale of data collection is appreciable, the analyses and the inferences drawn from them are lacking.

1). The survey data is known to contain all kinds of respondent biases, more so for such global surveys as done by this paper. This hurts the external validity of the findings in the paper.

2). The question of "tightness" of culture comes out of nowhere in the introduction section of the paper. Of all the important questions that one could study about COVID-19, it's impact on the tightness of culture via the lens of tightness-looseness theory seems to be the most contrived and also the least impactful. The paper needs to have a much clearer motivation for studying this research question.

3). (This is perhaps my most important concern.) The paper contains two types of empirical analyses. The first one is a set of multi-level models and the second is the difference-in-difference analysis. Neither the main paper or the SI contains any equations that clearly describe the the Y and the X variables as is customary in any paper using these analyses. The most important aspect of the paper (the empirical analyses) are described in passing in the main body of the paper where the authors just describe what the Y variables are.

I have serious concerns about the diff-in-diff analysis. Diff-in-diff relies on several key assumptions, e.g., parallel trends and the availability of a matched control group. The paper never shows any parallel trends plots or discusses what the control and treated groups are in the quasi-

experimental setup that they have. Does the diff-in-diff measure differences across countries before and during COVID-19 (via the "before" and "during" surveys)? What are the various variables that are controlled for in the diff-in-diff regression? The only thing that I can see are the two tables S3 and S12 which discuss the diff-in-diff results.

4). The second survey was conducted in the early stages of COVID-19 (april-june 2020), so even if a rigorous empirical analysis were to be done addressing the concerns I raised above, it is unclear whether those results would generalize or not since they're not "equilibrium" findings. The early stages of COVID-19 were tumultuous and most social behaviors in that phase would not generalize to the steady-state of the pandemic when things "settled down."

Reviewer #4:

Remarks to the Author:

This is a valuable study that exploits a powerful and (lets hope) unique worldwide shock. It addresses some open questions: Is the effect of threat on tightness a matter of slow changes to institutions or fast individual responses? Is it domain general or just to certain kinds of norms or norms relevant to the threat. This evidence suggests that there is short term reaction only to the norms that are relevant -- consistent with a rational adjustment of behavior. I like that the authors are intellectually modest about what can be concluded.

Reviewer #5:

Remarks to the Author:

It is an interesting paper. My main concern is the premise. I do not see a compelling argument why Covid might alter social norms or culture. Covid as consequential it was and is perceived largely as a "public health" issue and its societal response and reaction has been marked by fear, and panic. Not quite reason or a systematic process which might lead to altering social norms or culture. Say introduction of iPhones may have been.

REVIEWER 1

#	Question/Comment	Response
1	This paper tests the predicts of cultural tightness-looseness theory by seeing how norms changed in the early months of the COVID pandemic. It takes advantage of an earlier published experiment on cross-cultural variation in norms, and sees how those norms (and enforcement thereof) differed between the original data collection (up to late 2019) versus April-June 2020 when many countries were first closing down due to COVID. They find that contrary to expectations, norms became *less* tight during the pandemic (except for handwashing for obvious reasons). This speaks against some of the predictions of cultural tightness-looseness theory. Overall, this was an interesting paper that was a pleasure to read. Plus, it provides an important test of an influential theory using a global natural experiment and a rich dataset. Overall I think the quality and broad interest make it a good candidate for publication in Nature Communications.	We appreciate Reviewer 1's comments.
2	My biggest question is whether these data are best looked at in April to June 2020, and whether there would have been sufficient time for cultural change to occur given that the pandemic really started in March. Ideally they would have also collected data a little later too, such as late 2020 or early 2021, to allow for sufficient cultural change to have occurred and test three time points (pre-pandemic, early pandemic, later pandemic). However, I recognize that that is no longer a possibility, so the question is really whether cultural change would have occurred within the first few months of the pandemic. If there had been no change in norms (or enforcement thereof), the results would have been inconclusive. However, the results are significant in the opposite direction than predicted, which is at least evidence against a simple "any threat leads to general tightening" explanation. As such, I don't think it's a major problem that there are only data from the early pandemic. However, this does require more discussion to justify whether norm change could occur on such short timescales. There is some discussion, but the authors should do more to justify the timescale they looked at, and what it might have looked at had they collected the second wave a few months later to allow for more time for cultural change.	We agree with Reviewer 1 that a later measure is also important but think that this is a separate, and equally important, outcome to study. Indeed, we specifically decided to focus on the short-run consequences on the COVID-19 pandemic in this study. In our view, there are five reasons why this was a good choice: three substantive and two methodological. (1) Most disruption occurred in the months right after the emergence of the pandemic¹. This extremity provided a strong "treatment" and thus a good test of the theory. (2) Early data give insights into the non-equilibrium dynamics of how cultures move from one stable state to another. Some of the changes we observe may settle into equilibria in the long run while others may not. Whatever outcome ultimately holds, knowing the intermediate steps is a crucial part of further theory building. (3) Theories of norm and cultural change tend to be broad about the amount of time needed for such changes to take place²⁻⁴. While most suggest long-term change, this is not entirely clear as various caveats are introduced which would allow for short-term change in response to systemic threats (e.g. p. 23 in⁴). Indeed, there is an ongoing debate in about whether such change happens intra-generationally or inter-generationally⁵. With

		our findings, we delimit the scope of norm change theories providing evidence that longer time periods are needed for change to occur even in the face of dramatic collective threats. (4) Early Wave 2 data collection reduces endogeneity concerns. It does so by decreasing the probability that other large-scale crises (e.g. an economic crash) affect these cultures and confounding the effects we find. Additionally, it reduces concern from other confounds like differences in GDP growth or healthcare infrastructure development (both of which would could affect norms⁴) that would arise in longer-term measures. (5) We anticipated that other researchers would not be in a position to collect early data, and have pre-pandemic comparison data, and this would make our dataset rare, to our knowledge unique on the topic. In contrast, long term data on cultural traits (such as trust, respect for authority, national pride, but not tightness-looseness or the norm measures we have) will be collected in established surveys (e.g., the World Value Survey and the European Value Study). But they will lack a perspective on short-term effects. A final important reason is that even if we had decided to collect data a few months later, the same objection could have been raised. There is no easy way to identify when we would have waited long enough to allow for cultural change. We thus decided to focus on the short-run consequence and allow future work to investigate the long-term consequences. We include our reasoning in the manuscript on (p. 5, ll. 165-176) while also highlighting that our timeframe prevents us from studying the long-run consequences.
3	I have two further minor comments that are easily resolved. First, in some places in the supplementary material it says they “disposed of” some variables. Does this mean that they excluded these variables? If so, this requires justification. Or is this just a funny way of saying that they use those variables, perhaps a translation error?	We thank the reviewer for pointing out this mistake. We have now corrected the statement (which is now included in “Methods, Metanorm scenarios”, pp. 10-11).
4	Second, in table S5 it should clarify what exactly the dependant variable is, e.g., the appropriateness of that action.	We thank the reviewer for this comment. We now fully described the dependent variables in the description of tables. We have also made further

		changes to improve the clarity of all the supplementary table descriptions.
--	--	---

REVIEWER 2

#	Question/Comment	Answer
5	This paper tests whether, early in the COVID pandemic, the general tightness of societies increased and whether the strength of specific norms increased. As they report, in many cases there was no effect of the COVID pandemic. The two exceptions were that hand hygiene norms got stronger (which is relevant in general to transmissible diseases) and, surprisingly, the samples studied tended to get overall looser in their approach to norms. I think this is a strong paper. It asks several related, clear questions. It has a huge sample, with people from throughout the world. It is pre-registered. I also appreciate that the authors of the study are working to publish findings that, while not contradicting their theory, are nonetheless perhaps a bit uneasy for their theory to account for.	We appreciate the reviewers' comments.
6	The only major concern I had with the project was the standardization: The tightness-looseness items are standardized based on ratings from 50 "appropriate" questions. Why is this necessary? No explanation is given. Instead, the reader is referred to an earlier paper. But I didn't follow the conceptual rationale in that paper either; primarily, the standardization section of that paper also just describes what was mechanically done. Since this is one of the main dependent variables, readers should fully understand why it was calculated as it was. Personally, I think it's a bit strange to directly subtract scales from each other when the anchors are different. I will note that I do not have expertise in multilevel regression, so I can not assess the appropriateness of their modeling choices. I support publication with revisions.	Standardization is needed to adjust for cross-cultural variation in response sets given that some cultures are more likely to provide extreme responses or acquiesce to survey items than others^{3,6}. We follow guidelines from cross-cultural psychology⁶⁻⁸ and calculate appropriateness scores by averaging each individual's responses to 50 appropriateness items that all used the same response scale, from extremely inappropriate to extremely appropriate. This score is then subtracted from subjects' responses in the tightness-looseness questionnaire (6 items from³). The final individual TL scores are computed by averaging the adjusted 6 items. Nonetheless, we certainly understand why the reviewer would consider it strange to subtract scales from each other (and appreciate too that other readers may feel the same). So, we also checked the correlation between unstandardized and standardized TL scores, which is very high ($r=0.84$ for Wave 1 measures, $r=0.85$ for Wave 2 measures; in both cases $p<0.001$) and checked if our results change with unstandardized TL scores. We find they do not. We have added the above to the Methods section (p. 10, "Tightness-Looseness scores").
7	Minor typo: in the "meta norms and punishing" section in the results, you should refer to Fig 1C, not 1D.	Thank you for pointing this out. It is now corrected.

REVIEWER 3

#	Question/Comment	Response
8	This paper uses two large global (43 countries) surveys to understand the impact of a global shock as big as COVID-19 on social norms and behaviors. The paper reports the results of surveying >30,000 people before (April-Dec 2019) and in the initial phase (April-June 2020) of COVID-19. The surveys contained various questions about the social norms. The paper uses tightness-looseness theory and the data from the two surveys to study the change of social norms. A difference-in-difference analysis is also done using the difference in the responses from the two surveys as the dependent variable. The results show that COVID-19 strengthened hand-washing norms but did not change most other norms. The paper is well written and reports on a highly impressive large-scale data analysis on the societal impacts of COVID-19. Kudos to the entire team of authors on writing this paper! While the scale of data collection is appreciable, the analyses and the inferences drawn from them are lacking.	We appreciate the response. We also agree that the description of our analyses was lacking; we have added a methods section at the end of the manuscript that we believe addresses the reviewer's concerns.
9	1). The survey data is known to contain all kinds of respondent biases, more so for such global surveys as done by this paper. This hurts the external validity of the findings in the paper.	We agree that survey data are known to be afflicted by multiple kinds of bias, e.g. social desirability. However, these are general challenges faced by the survey literature and not specific to our paper. Perhaps the reviewer had a more specific point in mind concerning cross-cultural surveys. One of the issues known to occur is variation in systematically providing extreme or intermediate answers to survey questions⁶. This kind of clustering in responses would not reflect true cultural variation in the target construct (e.g. Tightness-Looseness) but rather a general response to answering questions. To account for this, we use an approach known as response-set standardization (see Methods). Our results are also robust without response-set standardization (see Methods, Tightness-Looseness scores). Additionally, the questions we use have been previously utilized by other surveys and are found to have convergent validity as well as other relevant properties. Tightness-Looseness questions have been found to predict a wide range of social, economic, and political factors^{3,9,10}. Ranging from innovation to

		criminal justice policy and COVID-19 response. Our own findings also point to the reliability of the Tightness-Looseness measures. Consider that our 2020 data have a correlation of $r=0.85$ (see also Fig S4) with data collected by Gelfand and co-authors³ nearly twenty years before in 2000-2003. Metanorms have been recently shown in a 56 country dataset (which is the source of our Wave 1 data) to be associated with various factors including punishing frequency, power distance, individualism, individual autonomy, emancipative moral judgments, tightness, national levels of gender equality, and median income⁷. Our other key items concerning situation-specific norms^{3,11} and handwashing norms^{12,13}, have likewise shown their validity.
10	2). The question of "tightness" of culture comes out of nowhere in the introduction section of the paper. Of all the important questions that one could study about COVID-19, it's impact on the tightness of culture via the lens of tightness-looseness theory seems to be the most contrived and also the least impactful. The paper needs to have a much clearer motivation for studying this research question.	Tightness-looseness is a key theory for understanding cross-cultural variation. Consider that Gelfand et al.'s 2011 paper—the foundational paper for the modern version of the theory—has been cited over 2200 times (according to Google Scholar as of March 2022). Myriad papers using this theory have found an important role for tightness-looseness across domains and fields including but not limited to: government, media openness, regulation strictness, criminal justice institutions (e.g. incarceration and criminal punishments), religious practices and beliefs, collective action, creativity, metanorms, constraints in daily life, happiness, urbanization, debt, health, tolerance for LGBT communities, gender equality, drug and alcohol use, and prejudice^{3,7,10,14-18}. Tightness-Looseness theory also has practical importance. For example, the theory is relevant to COVID-19 prevention. In the highly influential cross-disciplinary Nature Human Behaviour paper on the role of behavioral science in the COVID-19 pandemic response¹⁹, culture has its own subsection in which tightness-looseness theory is a key focus (pp. 463-464). Moreover, in recent work, Gelfand and co-authors find evidence for a relationship between tightness-looseness and pandemic deaths and cases during the first year of the pandemic⁹. Put differently, examining the effect that COVID-19 had on the tightness of cultures in the short term

		can help not only to predict future response of countries to similar situations and identify effective interventions to deal with these crises but also to better anticipate social changes that can impact our societies for generations to come. In sum, tightness-looseness theory is both scientifically influential and practically consequential and thus testing a key proposition of the theory is natural and impactful. It is also important to note that testing this theory was already our aim in March 2020, before collecting any data, and this can be verified from our initial pre-registration (see Methods, Study preregistration). We now mention additional correlates of tightness-looseness, have expanded the citation supporting these claims (p. 4, ll. 134-135), and highlight the importance of tightness-looseness (pp. 4, 5, ll. 151-154).
11	3). (This is perhaps my most important concern.) The paper contains two types of empirical analyses. The first one is a set of multi-level models and the second is the difference-in-difference analysis. Neither the main paper or the SI contains any equations that clearly describe the the Y and the X variables as is customary in any paper using these analyses. The most important aspect of the paper (the empirical analyses) are described in passing in the main body of the paper where the authors just describe what the Y variables are.	Thank you for this important comment. We have now added a methods section (p. 9 onward) that specifies our measures, the study preregistration, and all models.
12	I have serious concerns about the diff-in-diff analysis. Diff-in-diff relies on several key assumptions, e.g., parallel trends and the availability of a matched control group. The paper never shows any parallel trends plots or discusses what the control and treated groups are in the quasi-experimental setup that they have. Does the diff-in-diff measure differences across countries before and during COVID-19 (via the "before" and "during" surveys)? What are the various variables that are controlled for in the diff-in-diff regression? The only thing that I can see are the two tables S3 and S12 which discuss the diff-in-diff results.	Thank you for this very important comment which we took seriously and have taken multiple steps to address. The classic difference-in-differences (DiD) setup contains two groups of observations at two points in time: before an event and after an event. In this configuration, one set of observations is affected by the event (the treatment group) while the other is not (the control group). By comparing the difference in the differences between these groups, we can identify the average treatment effect on the treated (ATT)^{20,21}. While our approach is very similar in concept, it is not strictly a DiD. Specifically, it differs in two related ways. First, we have no totally untreated comparison group. All countries, to some extent,

were affected by the emergence of COVID-19. Even in places with near zero-cases, due to the interconnected nature of the modern world, simply observing other countries' experiences may have shaped the social norms of people within those unaffected countries. Thus, we have no (and think that it is not even possible) to have a realistic counterfactual untreated group. Second, instead of a treated and untreated group, we have many groups with different COVID-19 exposure levels.

The combination of these factors introduces complexities for identifying ATT that are still under development in the econometrics literature²². To account for this, we have undertaken a number of steps:

(1) We no longer mention that we use a DiD approach (although mention that it is similar in spirit; p. 6, ll. 215-220), and instead explicitly write that we exploit variation in our mechanisms measures (e.g. p. 6, ll. 215-220).

(2) We no longer claim to identify causal effects (i.e. ATT). Instead, we concretely describe the factors that our approach helps with. Specifically, the before and after comparison of Wave 1 to Wave 2 excludes the possibility of reverse causality and time-constant confounders while the mechanisms analysis (which exploits heterogeneity in COVID-19 intensity) reduces further potential confounds and allows us to shed light on the association between our proposed factors affecting the change in cultural measures included in our study (p. 6, ll. 215-223). There are multiple well-known papers that take this approach²³⁻²⁶.

(3) Nevertheless, to alleviate concerns of pre-pandemic differences in tightness-looseness trends, we check how this has changed by comparing data from³ and⁷. We find evidence consistent with the parallel trends assumptions of DiD approaches. That is, changes in TL across countries pre-pandemic were independent of their TL scores. More specifically, TL was highly stable across these two time periods. We describe this when outlining the tightness-looseness scores (p. 10, ll. 392-405).

To address the point about variables, we include the full model specifications in the Methods including controls. Specifically, we control for mean age, % female, and % students.

13	4). The second survey was conducted in the early stages of COVID-19 (april-june 2020), so even if a rigorous empirical analysis were to be done addressing the concerns I raised above, it is unclear whether those results would generalize or not since they're not "equilibrium" findings. The early stages of COVID-19 were tumultuous and most social behaviors in that phase would not generalize to the steady-state of the pandemic when things "settled down."	We believe that we learn a lot of important knowledge from studying the effect of the COVID-19 pandemic in the months directly after its emergence. Conceptually, it gives us insights into the non-equilibrium dynamics of how cultures move from one stable state to another. Some of the changes we observe may settle into equilibrium in the long run while others may not. Whichever outcome ultimately holds, knowing the intermediate steps is a crucial part of theory building. As an analogy, much modern social science is not only about causal effects (does X cause Z?), but also about the causal pathways, e.g. Y, through which X shapes Z ($X \rightarrow Y \rightarrow Z$). Likewise, we study the intermediate step before norms settle into stable states. Additionally, see our responses to comment #2 which address a closely related point. We include a section discussing this point in the manuscript (p. 5, ll. 165-176).
----	--	--

REVIEWER 4

#	Question/Comment	Response
14	This is a valuable study that exploits a powerful and (lets hope) unique worldwide shock. It addresses some open questions: Is the effect of threat on tightness is a matter of slow changes to institutions or fast individual responses? Is it domain general or just to certain kinds of norms or norms relevant to the threat. This evidence suggests that there is short term reaction only to the norms that are relevant -- consistent with a rational adjustment of behavior. I like that the authors are intellectually modest about what can be concluded.	We appreciate the reviewer's comments.

REVIEWER 5

#	Question/Comment	Response
15	It is an interesting paper. My main concern is the premise. I do not see a compelling argument why Covid might alter social norms or culture. Covid as consequential it was and is perceived largely as a "public health" issue and its societal response and reaction has been marked by fear, and panic. Not quite reason or a systematic process which might lead to altering social norms or culture. Say introduction of iPhones may have been.	COVID-19's emergence was extremely destabilizing event, breaking habits, behavioral patterns, and expectations. This shaped myriad behaviors: from how and where we work, how we meet and great people, how we travel, where we eat, how we exercise, and where we pursue leisure activities. Crucially, this occurred within whole populations at a similar time creating a shared event, as "coordinated disturbance", leading to a correlated upheaval and allowing the emergence of new behavioral regularities. Such behavioral regularities a foundational component of social norms²⁷⁻²⁹. Only with behavioral regularities can coordinated expectations about what others do arise, and this is an essential pre-cursor to inferring normativity to the behavioral regularity and thus ultimately the emergence of a social norm. Since new behavioral regularities emerged, it follows that they may have turned into new social norms. Multiple theories also predict that threats, in this case COVID-19, shape norms and cultures. Tightness-Looseness directly predicts that social norms become stronger in response to ecological threat³. Evolutionary Modernization Theory⁴, Parasite Stress Theory³⁰ and the broader notion of the behavioral immune system² all make broadly similar predictions. As an indication of how widely people believe that COVID-19 changes culture, consider that in a survey of behavioral scientists and lay Americans predicted that 10 of 15 dimensions—including explicit prejudice, implicit prejudice, traditionalism, generalized trust—will change³¹. Additionally, emerging cross-sectional studies find some evidence for attitudinal, value, and cultural change^{32,33}. However, almost all, with very few exceptions²⁴, use data collected only after the pandemic making it difficult to know the true change. In this regard, our work is unique as it is able to provide a before and after comparison

		of multiple norm variables in a large-scale and cross-country dataset.
--	--	--

REFERENCES

1. The Economist. Our normalcy index shows life is halfway back to pre-covid norms. *The Economist* <https://www.economist.com/graphic-detail/2021/07/03/our-normalcy-index-shows-life-is-halfway-back-to-pre-covid-norms> (2021).
2. Schaller, M. & Park, J. H. The Behavioral Immune System (and Why It Matters). *Curr Dir Psychol Sci* **20**, 99–103 (2011).
3. Gelfand, M. J. *et al.* Differences between tight and loose cultures: A 33-nation study. *Science* **332**, 1100–1104 (2011).
4. Inglehart, R. F. *Cultural Evolution: People's Motivations are Changing, and Reshaping the World*. (Cambridge University Press, 2018). doi:10.1017/9781108613880.
5. Kiley, K. & Vaisey, S. Measuring Stability and Change in Personal Culture Using Panel Data. *Am Sociol Rev* **85**, 477–506 (2020).
6. Gelfand, M. J., Raver, J. L. & Ehrhart, K. H. Methodological issues in cross-cultural organizational research. in *Handbook of research methods in industrial and organizational psychology* 216–246 (Blackwell Publishing, 2002).
7. Eriksson, K. *et al.* Perceptions of the appropriate response to norm violation in 57 societies. *Nature Communications* **12**, 1481 (2021).
8. van de Vijver, F. J. R. & Leung, K. *Methods and data analysis for cross-cultural research*. xiii, 186 (Sage Publications, Inc, 1997).
9. Gelfand, M. J. *et al.* The relationship between cultural tightness–looseness and COVID-19 cases and deaths: a global analysis. *The Lancet Planetary Health* **5**, e135–e144 (2021).
10. Chua, R. Y. J., Huang, K. G. & Jin, M. Mapping cultural tightness and its links to innovation, urbanization, and happiness across 31 provinces in China. *PNAS* **116**, 6720–6725 (2019).
11. Price, R. H. & Bouffard, D. L. Behavioral appropriateness and situational constraint as dimensions of social behavior. *Journal of Personality and Social Psychology* **30**, 579–586 (1974).
12. Eriksson, K., Dickins, T. E. & Strimling, P. Hygiene Norms Across 56 Nations are Predicted by Self-Control Values and Disease Threat. *Current Research in Ecological and Social Psychology* **2**, 100013 (2021).
13. Strimling, P., de Barra, M. & Eriksson, K. Asymmetries in punishment propensity may drive the civilizing process. *Nature Human Behaviour* **2**, 148–155 (2018).
14. Chua, R. Y. J., Roth, Y. & Lemoine, J.-F. The Impact of Culture on Creativity: How Cultural Tightness and Cultural Distance Affect Global Innovation Crowdsourcing Work. *Administrative Science Quarterly* **60**, 189–227 (2015).
15. Harrington, J. R. & Gelfand, M. J. Tightness–looseness across the 50 united states. *PNAS* **111**, 7990–7995 (2014).
16. Jackson, J. C., Gelfand, M., De, S. & Fox, A. The loosening of American culture over 200 years is associated with a creativity–order trade-off. *Nat Hum Behav* **3**, 244–250 (2019).
17. Jackson, J. C., Gelfand, M. & Ember, C. R. A global analysis of cultural tightness in non-industrial societies. *Proceedings of the Royal Society B: Biological Sciences* **287**, 20201036 (2020).
18. Jackson, J. C. *et al.* Ecological and cultural factors underlying the global distribution of prejudice. *PLOS ONE* **14**, e0221953 (2019).
19. Bavel, J. J. V. *et al.* Using social and behavioural science to support COVID-19 pandemic response. *Nature Human Behaviour* **4**, 460–471 (2020).
20. Angrist, J. D. & Pischke, J.-S. *Mostly harmless econometrics: An empiricist's companion*. (Princeton University Press, 2009).
21. Cunningham, S. *Causal Inference: The Mixtape*. (Yale University Press, 2021).
22. Callaway, B., Goodman-Bacon, A. & Sant'Anna, P. H. C. Difference-in-Differences with a Continuous Treatment. *arXiv:2107.02637 [econ]* (2021).
23. Nunn, N. & Qian, N. The Potato's Contribution to Population and Urbanization: Evidence From A Historical Experiment*. *The Quarterly Journal of Economics* **126**, 593–650 (2011).

24. Druckman, J. N., Klar, S., Krupnikov, Y., Levendusky, M. & Ryan, J. B. Affective polarization, local contexts and public opinion in America. *Nat Hum Behav* **5**, 28–38 (2021).
25. Finkelstein, A. The Aggregate Effects of Health Insurance: Evidence from the Introduction of Medicare*. *The Quarterly Journal of Economics* **122**, 1–37 (2007).
26. Acemoglu, D., Autor, D. H. & Lyle, D. Women, War, and Wages: The Effect of Female Labor Supply on the Wage Structure at Midcentury. *Journal of Political Economy* **112**, 497–551 (2004).
27. Bicchieri, C. *The grammar of society: The nature and dynamics of social norms*. (Cambridge University Press, 2006).
28. Bicchieri, C. & Mercier, H. Norms and beliefs: how change occurs. in *The complexity of social norms* (eds. Xenitidou, M. & Edmonds, B.) 37–54 (Springer, 2014).
29. Bicchieri, C. & Xiao, E. Do the right thing: but only if others do so. *J. Behav. Decis. Making* **22**, 191–208 (2009).
30. Fincher, C. L., Thornhill, R., Murray, D. R. & Schaller, M. Pathogen prevalence predicts human cross-cultural variability in individualism/collectivism. *Proceedings of the Royal Society B: Biological Sciences* **275**, 1279–1285 (2008).
31. Hutcherson, C. *et al.* Estimating societal effects of COVID-19. (2021) doi:10.31234/osf.io/g8f9s.
32. Hensel, L. *et al.* Global Behaviors, Perceptions, and the Emergence of Social Norms at the Onset of the COVID-19 Pandemic. *Journal of Economic Behavior & Organization* **193**, 473–496 (2022).
33. Greenfield, P. M., Brown, G. & Du, H. Shifts in ecology, behavior, values, and relationships during the coronavirus pandemic: Survival threat, subsistence activities, conservation of resources, and interdependent families. *Current Research in Ecological and Social Psychology* **2**, 100017 (2021).

Reviewers' Comments:

Reviewer #1:

Remarks to the Author:

The authors have satisfactorily addressed my concerns. As such, I think this paper is publishable.

I just realized one additional explanation for why the results did not support tightness-looseness theory. It's possible that the nature of the threat matters: people may be more responsive to social threats like outgroups than to asocial threats like disease natural disasters. The original tightness/looseness theory might not distinguish between these types of threats, but there is both theoretical and empirical evidence that people react differently to these different kinds of threats (see references below). I recommend that the authors add a sentence or two suggesting this possibility. Apologies that I didn't think of this earlier.

Theoretical: Gavrilets, S. (2015). Collective action and the collaborative brain. *Journal of the Royal Society Interface*, 12, 20141067. <http://dx.doi.org/10.1098/rsif.2014.1067>

Empirical: Barclay, P., & Benard, S. (2020). The effects of social versus asocial threats on group cooperation and manipulation of perceived threats. *Evolutionary Human Sciences*, 2, e54. <http://dx.doi.org/10.1017/ehs.2020.48>.

Other than that minor addition, I think this paper is ready to go, and I look forward to citing it.

Reviewer #2:

Remarks to the Author:

The authors' reply answered my question. I'm glad the choice about standardization ultimately does not matter much. I would support accepting this article.

Reviewer #6:

Remarks to the Author:

This paper tests how norms changed in the early months of the COVID pandemic using large scale survey data recorded before and recently after the emergence of COVID-19. It is an interesting work and I enjoy reading it the manuscript. It is well written and technically sounds. Therefore, I recommend publication.

REVIEWER 1

#	Question/Comment	Response
1	The authors have satisfactorily addressed my concerns. As such, I think this paper is publishable. I just realized one additional explanation for why the results did not support tightness-looseness theory. It's possible that the nature of the threat matters: people may be more responsive to social threats like outgroups than to asocial threats like disease natural disasters. The original tightness/looseness theory might not distinguish between these types of threats, but there is both theoretical and empirical evidence that people react differently to these different kinds of threats (see references below). I recommend that the authors add a sentence or two suggesting this possibility. Apologies that I didn't think of this earlier. Theoretical: Gavrilets, S. (2015). Collective action and the collaborative brain. Journal of the Royal Society Interface, 12, 20141067. http://dx.doi.org/10.1098/rsif.2014.1067 Empirical: Barclay, P., & Benard, S. (2020). The effects of social versus asocial threats on group cooperation and manipulation of perceived threats. Evolutionary Human Sciences, 2, e54. http://dx.doi.org/10.1017/ehs.2020.48. Other than that minor addition, I think this paper is ready to go, and I look forward to citing it.	We thank the reviewer for this feedback and have added the point and the references (p. 9, l. 312).

Reviewers' Comments:

Reviewer #7:

Remarks to the Author:

Dear editor,

I have reviewed the manuscript "Changes in Social Norms During the Early Stages of the COVID-19 2 Pandemic Across 43 Countries". As requested by the editor, I have focused on the methodological details of the report, paying particular attention to the power analysis and equivalence tests results, and their relation the reported conclusions. In general, I find the reporting to be of satisfyingly high methodological quality, but note that this is based on a somewhat superficial review on my part.

There are three issues that I believe need to be addressed before the article can be accepted for publication. I also note a fourth issue but leave it to the editor to decide whether this is something the authors need to address.

First, the justification for the "smallest effect size of interest" (SESOI) is weak. Using conventions/benchmark values to define what constitutes the SESOI makes inferences based on SESOI arbitrary, because conventions for what constitutes a "small", "medium" and "large" effect are themselves usually completely arbitrary. The weakness of this type of justification has been pointed out in several places, including in reference 32 in the manuscript. It would be better to base SESOI on either theory (though this is almost never possible in psychology), on prior research, or on reference effect sizes relevant to the field of study. This would give equivalence test results a more concrete interpretation. A significant equivalence test result could then be interpreted as "smaller than effects reported in previous studies", "smaller than the effect of X on tightness-looseness", etc. As it stands, significant equivalence test results can only be interpreted as something like "smaller than some arbitrary threshold value". It is unclear to me why the arbitrary conventional value of $d=0.1$ should be considered a threshold for "practical relevance". In a study that examines short-term changes and readily admits that real social change may take longer to fully develop, is there not good reason to suspect that even interesting changes might be quite small in size?

Second, equivalence tests are incorrectly restricted to only non-significant (null) results. The authors report that "For all null results reported ... we performed equivalence tests". However, the SESOI is the SESOI, regardless of whether the null-hypothesis test was significant or not. If a SESOI is determined, equivalence tests should be conducted for all null-hypothesis tests, and all equivalent results should be considered too small to be of practical relevance. In the present report, several test results which are clearly statistically equivalent to $d < 0.1$ (the CI does not cross SESOI on either side) are nonetheless reported as "significant" and treated as practically relevant in the discussion section. The most obvious example is the decrease in punishing frequency. This is certainly treated as a meaningful effect in the discussion. However, the lower CI on the punishing frequency decrease is -0.05 – clearly smaller than the specified SESOI of 0.1 . This should mean that the authors take the effect to be of no practical importance! Now, the authors can of course argue that this effect is interesting and damn the SESOI. After all, this SESOI is based on an arbitrary convention. This is perfectly fine, but something needs to change. Either interpretation of currently significant but equivalent results must change, or the SESOI must be changed (or abandoned). In either case, equivalence tests must be applied to all tests when they are used to determine practical relevance.

Third, the report mentions a few cases of "marginally significant effects", which is a nonsensical concept. If the authors wish to treat p-values as evidence, they should opt for a likelihoodist or Bayesian approach to their analyses, drop all use of hypothesis tests, and report likelihoods or Bayesian posteriors instead. If the authors wish to stick with the Neyman-Pearson test framework, they should act as if all nonsignificant/equivalent findings are too small to be of interest (accepting the 5% type 1 error rate for this decision). Allowing for "marginal significance" just means that the reader no longer take the stated alpha level of 5% seriously.

Fourth, I am surprised that the issue of selection bias does not receive more attention in the discussion section. The authors note that "we use convenience samples (albeit both students and

non-students). While this is unlikely to have substantial implications on our estimates, since the samples are broadly similar between the waves, it should be kept in mind when generalizing our findings to the broader populations” but do not specify why it should be kept in mind, or what potential generalizability pitfalls might be. Given how likely it is that “college students primarily living in the capital” are different from the general population of adults, both in terms of social norms held and in terms of COVID-19 experiences, I think it would be relevant to discuss how this affects our ability to generalize in more detail.

Signed,
Peder M. Isager

REVIEWER 7

#	Question/Comment	Response
1	Dear editor, I have reviewed the manuscript “Changes in Social Norms During the Early Stages of the COVID-19 2 Pandemic Across 43 Countries”. As requested by the editor, I have focused on the methodological details of the report, paying particular attention to the power analysis and equivalence tests results, and their relation the reported conclusions. In general, I find the reporting to be of satisfyingly high methodological quality, but note that this is based on a somewhat superficial review on my part. There are three issues that I believe need to be addressed before the article can be accepted for publication. I also note a fourth issue but leave it to the editor to decide whether this is something the authors need to address.	We appreciate your review and comments concerning our manuscript. We believe that we have addressed all the four points that you raise.
2	First, the justification for the “smallest effect size of interest” (SESOI) is weak. Using conventions/benchmark values to define what constitutes the SESOI makes inferences based on SESOI arbitrary, because conventions for what constitutes a “small”, “medium” and “large” effect are themselves usually completely arbitrary. The weakness of this type of justification has been pointed out in several places, including in reference 32 in the manuscript. It would be better to base SESOI on either theory (though this is almost never possible in psychology), on prior research, or on reference effect sizes relevant to the field of study. This would give equivalence test results a more concrete interpretation. A significant equivalence test result could then be interpreted as “smaller than effects reported in previous studies”, “smaller than the effect of X on tightness-looseness”, etc. As it stands, significant equivalence test results can only be interpreted as something like “smaller than some arbitrary threshold value”. It is unclear to me why the arbitrary conventional value of $d=0.1$ should be considered a threshold for “practical relevance”. In a study that examines short-term changes and readily admits that real social change may take longer to fully develop, is there not good reason to suspect that even interesting changes might be quite small in size?	Thank you for this comment. We carefully reviewed again the recommendations in^{1,2} and have fundamentally updated how we set the SESOI. As there is no prior evidence that we could draw on to set the SESOI, we decided to adopt a different strategy. Specifically, we use the variation in the tightness-looseness of social norms between Wave 0³ (data collected in 2002/2003) and Wave 1⁴ (data collected in 2019). Since these measures were collected during a period in which no clear global threat, certainly not at the scale of COVID-19, occurred, we look to this as an indicator of natural variation. We find that, on average, countries move up or down nearly 2 places in the rank ordering of TL between these two waves. Given that natural variation from our measures implies changes of 2 ranks, we set the SESOI at one rank larger: 3 ranks. We then converted this rank change into the equivalent change for each of the social norm measures on their raw scales and conducted the TOST procedure using the coefficients and standard errors derived from the model estimates displayed in the main text and supplementary materials (see updated section “Equivalence tests” on p. 14).

3	Second, equivalence tests are incorrectly restricted to only non-significant (null) results. The authors report that “For all null results reported ... we performed equivalence tests”. However, the SESOI is the SESOI, regardless of whether the null- hypothesis test was significant or not. If a SESOI is determined, equivalence tests should be conducted for all null-hypothesis tests, and all equivalent results should be considered too small to be of practical relevance. In the present report, several test results which are clearly statistically equivalent to $d < 0.1$ (the CI does not cross SESOI on either side) are nonetheless reported as “significant” and treated as practically relevant in the discussion section. The most obvious example is the decrease in punishing frequency. This is certainly treated as a meaningful effect in the discussion. However, the lower CI on the punishing frequency decrease is -0.05 – clearly smaller than the specified SESOI of 0.1. This should mean that the authors take the effect to be of no practical importance! Now, the authors can of course argue that this effect is interesting and damn the SESOI. After all, this SESOI is based on an arbitrary convention. This is perfectly fine, but something needs to change. Either interpretation of currently significant but equivalent results must change, or the SESOI must be changed (or abandoned). In either case, equivalence tests must be applied to all tests when they are used to determine practical relevance.	We agree with the reviewer and have now corrected this. Our motivation for only conducting equivalence tests on the non-significant results was based on a limited reading of an earlier editorial comment that we should have interpreted more fully as also applying to significant findings. With the updated SESOI and implementation, we find that tightness-looseness, situation-specific norms, metanorms, and punishing frequency are statistically equivalent (i.e. reject the SESOI null hypothesis). While handwashing norms fall far above the upper SESOI equivalence bound. We now include equivalence testing in the analytic approach (p. 6), have added the equivalence tests outcomes in the results (pp. 6-8), and have adjusted the discussion to reflect the non-meaningful interpretation of the change in tightness looseness and punishing frequency (pp. 8-9).
4	Third, the report mentions a few cases of “marginally significant effects”, which is a nonsensical concept. If the authors wish to treat p-values as evidence, they should opt for a likelihoodist or Bayesian approach to their analyses, drop all use of hypothesis tests, and report likelihoods or Bayesian posteriors instead. If the authors wish to stick with the Neyman-Pearson test framework, they should act as if all nonsignificant/equivalent findings are too small to be of interest (accepting the 5% type 1 error rate for this decision). Allowing for “marginal significance” just means that the reader no longer take the stated alpha level of 5% seriously.	We have changed both mentions of marginally significant to variants of “significant at the 10% level” (p. 8). We wish to mention that we do not place any weight on these cases in our discussion but also do not want to unnecessarily draw an arbitrary “bright line” at $p=0.05$. As the American Statistical Association’s⁵ latest statement on p-values writes: “Scientific conclusions and business or policy decisions should not be based only on whether a p-value passes a specific threshold. Practices that reduce data analysis or scientific inference to mechanical ‘bright-line’ rules (such as ‘$p < 0.05$’) for justifying scientific claims or conclusions can lead to erroneous beliefs and poor decision making.”
5	Fourth, I am surprised that the issue of selection bias does not receive more attention in the discussion section. The authors note that “we use convenience samples (albeit both students and non-students). While this is unlikely to have substantial implications on our	We now mention why this should be kept in mind: “Specifically, it is possible that social norm change, or a lack thereof, occurred differently outside of cities and varied with socio-economic factors” (p. 9).

estimates, since the samples are broadly similar between the waves, it should be kept in mind when generalizing our findings to the broader populations” but do not specify why it should be kept in mind, or what potential generalizability pitfalls might be. Given how likely it is that “college students primarily living in the capital” are different from the general population of adults, both in terms of social norms held and in terms of COVID-19 experiences, I think it would be relevant to discuss how this affects our ability to generalize in more detail.

Signed,
Peder M. Isager

REFERENCES

1. Lakens, D. Equivalence Tests: A Practical Primer for t Tests, Correlations, and Meta-Analyses. *Social Psychological and Personality Science* **8**, 355–362 (2017).
2. Lakens, D., Scheel, A. M. & Isager, P. M. Equivalence Testing for Psychological Research: A Tutorial. *Advances in Methods and Practices in Psychological Science* **1**, 259–269 (2018).
3. Gelfand, M. J. *et al.* Differences between tight and loose cultures: A 33-nation study. *Science* **332**, 1100–1104 (2011).
4. Eriksson, K. *et al.* Perceptions of the appropriate response to norm violation in 57 societies. *Nature Communications* **12**, 1481 (2021).
5. Wasserstein, R. L. & Lazar, N. A. The ASA Statement on p-Values: Context, Process, and Purpose. *The American Statistician* **70**, 129–133 (2016).

Reviewers' Comments:

Reviewer #8:

Remarks to the Author:

I have been invited to review this paper with the specific task of evaluating the use of equivalence tests, picking up from a previous reviewer (Peder Isager) who unfortunately was unavailable for a re-review. First off, I agree with all of the previous reviewer's comments and suggestions, and I appreciate the authors' attempts to incorporate them in their revision. My evaluation is as follows:

1. I commend the authors' effort to find a SESOI that is genuinely 'of interest' and not default-based. Unfortunately, though, I think the new approach is not ideal: Setting a SESOI based on the average rank change of countries in TL from 2002/03 (Wave 0) to 2019 (Wave 1) effectively assumes that no meaningful changes in TL occurred in any country in this time frame. I think it is obvious that this is not a plausible assumption — the nearly two decades between these time points saw various wars, epidemics, significant changes in political leaderships and even political systems, a global financial crisis, and the advent of social media, to name just a few events that may affect cultural norms in important ways. From this angle, it is not clear to me that the average rank change of a country in this period reflects 'natural variation' rather than genuine changes that may well be of theoretical interest.

Finding a meaningful SESOI in this context is difficult, and I do not have an easy solution. One imperfect alternative would be to not use average rank change, but average change overall between Wave 0 and 1 (i.e., mean change across all countries). The reason I think this would be slightly better is that it is more principled and fits the current operationalisation of the research question better (which is concerned with average change across countries, not country-level changes). The reason it's still far from ideal is that it also reflects a very bold assumption: This SESOI would assume that *there was no meaningful change in TL across countries between Wave 0 and 1*. Put differently, the assumption would be that there was no meaningful global trend up or down in this period. I personally don't have enough subject expertise to judge if this assumption is too obviously wrong to be useful.

2. Regardless of which SESOI justification the authors choose, this justification must be made explicit in the main text, which currently only says whether or not a given change was 'practically meaningful'. Especially in cases like this one where there seems to be no straightforward way of setting a principled SESOI (i.e., there is no shared understanding of which effect sizes would be practically meaningful), readers need to clearly understand what the SESOI entails in order to interpret the equivalence test results.

A different way of putting this is that a SESOI is a reference frame. Rather than saying 'this effect was not practically meaningful ($\Delta = \dots, \dots$)', the results would be easier to interpret with a statement such as 'this effect was statistically smaller than the baseline change from 2002/03 to 2019 (...)'. Of course spelling this out in detail each time would be tedious, but it should be possible to find a solution where the relevant terms are clearly defined in the beginning of the Results section (i.e., spelled out at least once in enough detail for the reader to clearly understand what comparison is being made).

3. It is not entirely clear to me whether the authors used baseline change *in TL* to set the SESOI for all variables (i.e. metanorms, punishing frequency, and so on), or if the SESOI for each variable was based on baseline change in that variable. The Method section currently reads as if the former is the case. If so, this approach makes the results difficult to interpret because it is unclear how baseline change in TL relates to change in e.g. hand washing norms (essentially an 'apples and oranges' problem). Ideally, the SESOI for each variable should be set using a reference that is meaningful for the respective variable. Depending on the specific variables and how they are measured, it is possible that doing so would lead to very similar results to the current approach, but this is not a given.

4. As a more minor point for reporting the SESOI and the results in general, I found it difficult to interpret the effect sizes without having a clear idea of the scale and the distribution (of each variable). Unless I missed it (apologies if so), the eventually analysed scales (i.e., after the reported transformations) and descriptive statistics should be reported somewhere.

5. As the previous reviewer noted, all statistical significance tests should be accompanied with equivalence tests. The authors have added equivalence tests to all t-tests of mean change, but not to the regression analyses. I'm not aware of a software package that includes equivalence tests for regression, but such a solution is not necessary: Equivalence can be tested simply by determining if the 90% CI (for $\alpha = 5\%$) of the effect overlaps with the SESOI on at least one side. This poses another challenge, as now SESOIs should be set not only for the analysis of mean changes, but also for the analysis of mechanisms, as the authors call it. However, doing so is of vital importance, as I will explain in the following point.

6. I also fully share the previous reviewer's viewpoint that the concept of 'marginal significance' is not useful and should be avoided. In response, the authors changed their language from 'marginally significant' to 'significant at the 10% level' and justify this with a very reasonable ASA statement which says that important decisions should not solely hinge on whether a p-value meets a specific threshold. This is not sufficient to address the previous reviewer's point. As he pointed out, by doing this, the authors effectively communicate that they do not take their preregistered alpha level of 5% seriously.

Of course the authors are free to set their alpha to 10%, but this also means that they will draw false-positive conclusions twice as often. In this specific case, I strongly recommend against doing so: The study has an extremely large sample size and thus very high power even for very small effects. When studying a true effect with very high power, the distribution of p-values becomes extremely left-skewed, meaning that even within the significant range, 'larger' p-values (e.g., $p = .02 - p = .05$) become extremely rare. In fact, they become much rarer than the same p-values when there is no true effect (in this case the p-distribution is completely flat). A highly educational interactive app illustrating this can be found here: <https://rpsychologist.com/d3/pdist/>

What this means is that with very high power for a hypothesised effect, a relatively large but significant p-value (e.g., $p = .04$) will be more likely to occur when there is no true effect. This seemingly paradoxical fact is also known as Lindley's paradox. Translated to the present study, this means that increasing the alpha level is very unwise, because it will lead to a disproportionate increase of false-positive results and provide virtually no benefit in terms of decreasing false-negative results.

To circle back to the previous point, the issue may become much more clear if we consider the size of the specific effect in question. The authors insist that there is 'some indication (significant at the 10% level) that frequency of confronting also decreased more in countries with higher governmental stringency ($b = -0.002$, 95% CI = $[-0.004; 0.000]$, $p = 0.099$).' In other words, the authors argue that an effect size of $b = -0.002$ (with the extreme end of the 95% CI being -0.004) is potentially meaningful. I personally think that this is a difficult case to make. But it will become much more clear with an equivalence test against a SESOI based on a principled consideration of what effect sizes should be too small to matter.

7. All preregistered hypotheses predict positive relationships. Strictly speaking, this means that the authors should only interpret the results with regard to whether there was or wasn't a positive change. I personally think that the current two-sided interpretation is fine (doing so does not inflate the type-I error rate beyond 5%), but wanted to point this out because it could be used to simplify the reporting.

8. The authors repeatedly refer to whether effects were 'robust' or not. The meaning of this term should be clearly defined in the main text to avoid misinterpretations.

9. Finally, although my review is focused on the use of statistical inference in this paper, I also have two more general remarks about the theoretical background of the tested hypotheses:

First, I found it surprising that the authors do not use direct measures of disease severity to test their main hypothesis (that COVID-19 caused cultural tightening) although these measures clearly seem to be an asset for capturing the research question in a more precise way (countries with higher vs lower disease severity should have tightened more vs less, respectively, and such differences 'wash out' as noise when looking only at mean change pre vs post COVID).

Second, the authors use proxy measures for disease severity (perceived prevalence, perceived threat, and government stringency) to study the mechanism underlying cultural tightening in response to COVID-19. If I understand correctly, the theoretical claim here is that actual disease severity has a causal effect on the proxy measures and the proxy measures have a causal effect on cultural tightness. But to me it is not clear that the proxy measures are a) theoretically separate from the concept of cultural tightness and b) causing cultural tightness rather than being caused by it. To the first point, governmental stringency very clearly seems to fall into the conceptual definition of cultural tightness. To the second point, it seems quite plausible that cultural tightness may causally affect perceived threat — one might even think that this is one of the functions of cultural tightness (i.e., tightening may serve to reduce the threat and consequently reduce perceived threat/fear). In any case, the relationships between these variables seem rather complicated and the authors' analysis currently does not justify causal inferences. I would thus recommend to make it clear that any such inferences (about the 'mechanism') rest on assumptions rather than empirical evidence.

Sincerely,

Anne Scheel

REVIEWER 8

#	Question/Comment	Response
1	I have been invited to review this paper with the specific task of evaluating the use of equivalence tests, picking up from a previous reviewer (Peder Isager) who unfortunately was unavailable for a re-review. First off, I agree with all of the previous reviewer's comments and suggestions, and I appreciate the authors' attempts to incorporate them in their revision. My evaluation is as follows:	Thank you for your thorough review.
2	1. I commend the authors' effort to find a SESOI that is genuinely 'of interest' and not default-based. Unfortunately, though, I think the new approach is not ideal: Setting a SESOI based on the average rank change of countries in TL from 2002/03 (Wave 0) to 2019 (Wave 1) effectively assumes that no meaningful changes in TL occurred in any country in this time frame. I think it is obvious that this is not a plausible assumption — the nearly two decades between these time points saw various wars, epidemics, significant changes in political leaderships and even political systems, a global financial crisis, and the advent of social media, to name just a few events that may affect cultural norms in important ways. From this angle, it is not clear to me that the average rank change of a country in this period reflects 'natural variation' rather than genuine changes that may well be of theoretical interest. Finding a meaningful SESOI in this context is difficult, and I do not have an easy solution. One imperfect alternative would be to not use average rank change, but average change overall between Wave 0 and 1 (i.e., mean change across all countries). The reason I think this would be slightly better is that it is more principled and fits the current operationalisation of the research question better (which is concerned with average change across countries, not country-level changes). The reason it's still far from ideal is that it also reflects a very bold assumption: This SESOI would assume that *there was no meaningful change in TL across countries between Wave 0 and 1*. Put differently, the assumption would be that there was no meaningful global trend up or down in this period. I personally don't have enough subject expertise to judge if this assumption is too obviously wrong to be useful.	Based on your comments, and further consideration, we agree that setting the SESOI based on average rank change is not ideal. Consequently, we have dropped this approach. Yet we also agree that using average change in tightness scores across countries is also not ideal. As, indeed you highlight, this would imply a bold assumption. Moreover, there would also be two further difficulties in using average score changes. First, Wave 0 tightness-looseness data, which we extract from the foundational 2011 publication on tightness-looseness, are only available in a transformed form¹ and already include individual-level standardization (based on the situation-specific norm scores) to account for cross-cultural variation in response sets. This means that we cannot cleanly identify the average Wave 0 to Wave 1 changes in tightness. (Note: for the tests of stability described in the Methods and shown in Figure S7, we do not need to separate the source of variation). Second, earlier data— analogous to Wave 0—on our other outcome variables does not exist. This means that we would need to adopt a different SESOI setting procedure for situation-specific norms, metanorms, punishing frequency, and hand washing norms. As a consequence, we also do not use the average change in scores across the countries. Instead, given that there is no ideal solution, we adopt an approach that is clear and consistently calculated for each measure. Specifically, we now set the SESOI in two ways: we compare the observed changes to (i) the within-country variation and (ii) between-country variation for each of our outcome measures. If the change across time in a country is negligible both

		compared to within- and between-country variation then it seems plausible that it is a negligible change. Concerning the SESOI for the analyses of mechanisms (as you highlight in comment #6), there is no existing evidence nor sensible substantive grounds on which to base it. Consequently, we have to use a benchmark approach and set the SESOI for these at $\beta = \pm 10$ based on². More generally, this difficulty and uncertainty in setting the SESOI highlights how problematic it can be to specify clear-cut borders that imply a relevant small effect size. This is particularly true in research like ours that has no agreed-upon way of setting the SESOI. Indeed, our work is the first large-scale study that is able to speak to social norm change in times of crisis and thus we cannot rely on precedent. We were also concerned about two additional factors during this process: (1) In contrast to the key components of our paper, the SESOI was not pre-registered and instead defined ex post. (2) Adding equivalence tests to every significance test in the results, including for the mechanisms, (as requested in comment #6) gives them an unwarranted prominence that is problematic given the aforementioned issues. Consequently, we now include separate sections on equivalence testing (pp. 6, 8) and in the Methods section (pp. 14-15). There we explain that (i) the SESOI was specified ex post in order to understand the null or small effects, (ii) that we use between and within country variation from Wave 1 for the between wave change, (iii) we use convention-based SESOI for the mechanism analyses (see response to comment #6 for details), and the (iv) limitations and issues with these. We have also updated the manuscript at the relevant sections to reflect these changes.
3	2. Regardless of which SESOI justification the authors choose, this justification must be made explicit in the main text, which currently only says whether or not a given change was 'practically meaningful'. Especially	We now explain the reasoning for choosing the SESOI, spell out exactly how we set them and why, and that whatever effect we find is smaller

	in cases like this one where there seems to be no straightforward way of setting a principled SESOI (i.e., there is no shared understanding of which effect sizes would be practically meaningful), readers need to clearly understand what the SESOI entails in order to interpret the equivalence test results. A different way of putting this is that a SESOI is a reference frame. Rather than saying 'this effect was not practically meaningful (delta = ..., ...)!', the results would be easier to interpret with a statement such as 'this effect was statistically smaller than the baseline change from 2002/03 to 2019 (...)'. Of course spelling this out in detail each time would be tedious, but it should be possible to find a solution where the relevant terms are clearly defined in the beginning of the Results section (i.e., spelled out at least once in enough detail for the reader to clearly understand what comparison is being made).	than (or not) the between and within country variation in the variable (pp. 6, 8, 14, 15).
4	3. It is not entirely clear to me whether the authors used baseline change *in TL* to set the SESOI for all variables (i.e. metanorms, punishing frequency, and so on), or if the SESOI for each variable was based on baseline change in that variable. The Method section currently reads as if the former is the case. If so, this approach makes the results difficult to interpret because it is unclear how baseline change in TL relates to change in e.g. hand washing norms (essentially an 'apples and oranges' problem). Ideally, the SESOI for each variable should be set using a reference that is meaningful for the respective variable. Depending on the specific variables and how they are measured, it is possible that doing so would lead to very similar results to the current approach, but this is not a given.	Given the lack of data (see response to comment #2), we had to use the average rank change for tightness-looseness also for the other variables. Since we now change our approach (as described in our response to comment #2), we now use the variation for each variable to separately set the SESOI.
5	4. As a more minor point for reporting the SESOI and the results in general, I found it difficult to interpret the effect sizes without having a clear idea of the scale and the distribution (of each variable). Unless I missed it (apologies if so), the eventually analysed scales (i.e., after the reported transformations) and descriptive statistics should be reported somewhere.	Thank you for noticing this. We have now added the relevant descriptive statistics in the Methods section (pp. 11-12).
6	5. As the previous reviewer noted, all statistical significance tests should be accompanied with equivalence tests. The authors have added equivalence tests to all t-tests of mean change, but not to the regression analyses. I'm not aware of a software package that includes equivalence tests for regression, but such a solution is not necessary: Equivalence can be tested simply by determining if the 90% CI (for	We did not previously have equivalence tests for the mechanisms predicting changes (i.e. Fear of COVID-19, Perceived COVID-19 prevalence, and Government Stringency Index) since there is no existing literature upon which we could base our SESOIs for these variables. Still, given the reviewer's request, we now also include equivalence tests for the mechanisms

alpha = 5%) of the effect overlaps with the SESOI on at least one side. This poses another challenge, as now SESOIs should be set not only for the analysis of mean changes, but also for the analysis of mechanisms, as the authors call it. However, doing so is of vital importance, as I will explain in the following point.	analyses. As there is no existing literature or objective criteria upon which to base the SESOI, we use a benchmark approach and set the SESOI at a standardised $\beta = \pm .10$, which Cohen² considers a small effect size.
7 6. I also fully share the previous reviewer's viewpoint that the concept of 'marginal significance' is not useful and should be avoided. In response, the authors changed their language from 'marginally significant' to 'significant at the 10% level' and justify this with a very reasonable ASA statement which says that important decisions should not solely hinge on whether a p-value meets a specific threshold. This is not sufficient to address the previous reviewer's point. As he pointed out, by doing this, the authors effectively communicate that they do not take their preregistered alpha level of 5% seriously. Of course the authors are free to set their alpha to 10%, but this also means that they will draw false-positive conclusions twice as often. In this specific case, I strongly recommend against doing so: The study has an extremely large sample size and thus very high power even for very small effects. When studying a true effect with very high power, the distribution of p-values becomes extremely left-skewed, meaning that even within the significant range, 'larger' p-values (e.g., $p = .02$ - $p = .05$) become extremely rare. In fact, they become much rarer than the same p-values when there is no true effect (in this case the p-distribution is completely flat). A highly educational interactive app illustrating this can be found https://rpsychologist.com/d3/pdist/ What this means is that with very high power for a hypothesised effect, a relatively large but significant p-value (e.g., $p = .04$) will be more likely to occur when there is no true effect. This seemingly paradoxical fact is also known as Lindley's paradox. Translated to the present study, this means that increasing the alpha level is very unwise, because it will lead to a disproportionate increase of false-positive results and provide virtually no benefit in terms of decreasing false-negative results. To circle back to the previous point, the issue may become much more clear if we consider the size of the	We now say not significant at the 5% level.

	specific effect in question. The authors insist that there is 'some indication (significant at the 10% level) that frequency of confronting also decreased more in countries with higher governmental stringency ($b = -0.002$, 95% CI=[-0.004; 0.000], $p = 0.099$).' In other words, the authors argue that an effect size of $b = -0.002$ (with the extreme end of the 95% CI being -0.004) is potentially meaningful. I personally think that this is a difficult case to make. But it will become much more clear with an equivalence test against a SESOI based on a principled consideration of what effect sizes should be too small to matter.	
8	7. All preregistered hypotheses predict positive relationships. Strictly speaking, this means that the authors should only interpret the results with regard to whether there was or wasn't a positive change. I personally think that the current two-sided interpretation is fine (doing so does not inflate the type-I error rate beyond 5%), but wanted to point this out because it could be used to simplify the reporting.	We pre-registered two-sided tests and stick to those.
9	8. The authors repeatedly refer to whether effects were 'robust' or not. The meaning of this term should be clearly defined in the main text to avoid misinterpretations.	We have expanded the description of what we mean by robust on p. 6, ll. 216-217.
10	9. Finally, although my review is focused on the use of statistical inference in this paper, I also have two more general remarks about the theoretical background of the tested hypotheses: First, I found it surprising that the authors do not use direct measures of disease severity to test their main hypothesis (that COVID-19 caused cultural tightening) although these measures clearly seem to be an asset for capturing the research question in a more precise way (countries with higher vs lower disease severity should have tightened more vs less, respectively, and such differences 'wash out' as noise when looking only at mean change pre vs post COVID). Second, the authors use proxy measures for disease severity (perceived prevalence, perceived threat, and government stringency) to study the mechanism underlying cultural tightening in response to COVID-19. If I understand correctly, the theoretical claim here is that actual disease severity has a causal effect on the proxy measures and the proxy measures have a causal effect on cultural tightness. But to me it is not clear that the proxy measures are a) theoretically separate from the concept of cultural tightness and b) causing cultural tightness rather than being caused by it. To the first	Thank you for these additional comments. Concerning the direct measures of disease severity: we do also use COVID-19 cases and deaths in alternative model specifications. They do not affect our results (see Methods and Supplementary Table S6). There are also good reasons to avoid focusing on cases and deaths, especially at the stage of the pandemic that we study (March-July 2020). Namely, enormous measurement issues. Testing practices, classification of deaths due to COVID-19 or not, as well as epidemiological reporting varied systematically from country to country due to available resources and public policy. Concerning the relationship between tightness-looseness and the mechanisms, this is an interesting point and remains a fascinating avenue for future work. Additionally, we do not claim causality and indeed there is no causal language in the paper.

point, governmental stringency very clearly seems to fall into the conceptual definition of cultural tightness. To the second point, it seems quite plausible that cultural tightness may causally affect perceived threat — one might even think that this is one of the functions of cultural tightness (i.e., tightening may serve to reduce the threat and consequently reduce perceived threat/fear). In any case, the relationships between these variables seem rather complicated and the authors' analysis currently does not justify causal inferences. I would thus recommend to make it clear that any such inferences (about the 'mechanism') rest on assumptions rather than empirical evidence.

REFERENCES

1. Gelfand, M. J. *et al.* Differences between tight and loose cultures: A 33-nation study. *Science* **332**, 1100–1104 (2011).
2. Cohen, J. *Statistical Power Analysis for the Behavioral Sciences*. (Routledge, 1988). doi:10.4324/9780203771587.

Reviewers' Comments:

Reviewer #8:

Remarks to the Author:

I would like to thank the authors for their considerate engagement with my review comments. Several, though not all, of my previous points have now been resolved. I will go through them in order of the most recent rebuttal letter.

1. My first comment concerned the justification for the SESOI of the equivalence tests for the central analysis (TL change between Wave 1 and Wave 2). In response, the authors changed their justification strategy and proposed two new SESOIs: the within-country and between-country variation (1 SD) of the outcome at Wave 1, set individually for TL, situation-specific norms, metanorms, and punishing frequency. I sincerely appreciate the authors' engagement with my arguments and their effort to provide a better solution. Unfortunately, though, I think that the new SESOI strategy is still not suitable (and may be less preferable than the previous version). First, the (within-country as well as between-country) variation of the outcome variable at Wave 1 is part of the effect size of the difference between Wave 1 and Wave 2. This means that the SESOI effectively depends on the effect size, which invalidates the equivalence test. For the test to be informative, the bound against which the effect is tested should be an independent criterion. Second, 1 standard deviation of the outcome variable is generally not a negligible effect size – it corresponds to a Cohen's d of 1. In many situations, finding that an effect is statistically smaller than $d = 1$ would be seen as not very informative.

I understand this search for a SESOI may feel frustrating, especially because my lack of expertise in this research area means that I cannot provide a worked-out solution myself. The most pragmatic approach I can see at the moment is for the authors to investigate whether there are any mean differences in the data and findings of Wave 0 (Gelfand et al., 2011) that may serve as a reasonable benchmark. The guiding principle would be to find a contrast that is deemed theoretically inconsequential/uninteresting (or merely on the border to theoretically consequential/interesting). If no suitable empirical SESOI can be identified, I would consider going back to a default value such as a 'small effect' according to Cohen. I realise that this approach had been criticised by a previous reviewer and therefore changed, but if there are genuinely no other means of identifying a SESOI that has meaning for the specific research area and target theory, I believe that defaults can be used as a last option. If this option must be used, I think it would be worth for the authors to note this situation explicitly, because understanding and building agreement about which effect sizes are meaningful is an important task for the entire research community in a field.

2. The authors have implemented my request to clearly explain the justification for the SESOI in the text. My only remaining comment about this concerns the reporting of the results of the equivalence tests, in which tests results that are not equivalent are called 'practically meaningful'. I recommend against this phrasing: Whether or not an effect is seen as practically meaningful is not a matter of the outcome of an equivalence test per se, but depends on the SESOI. If the SESOI is not (practically) meaningful, the test result also doesn't provide such information. With this I just want to clarify that 'practically meaningful' is not a technical term like 'statistically significant' and should not be used mechanically.

3. The authors have also implemented my recommendation to set individual SESOIs for the tests of change in TL, situation-specific norms, metanorms, and punishing frequency. As I explained above, unfortunately I don't think that the new strategy for these tests is adequate. Ideally, my suggestion above could lead to finding meaningful SESOIs of each outcome variable. If this is impossible, a blanket SESOI will have to do (keeping the limitation of the apples vs oranges problem in mind).

4. Thank you for adding the descriptive statistics.

5. The authors have now added equivalence tests for the 'mechanism' analyses, with a default SESOI of $\beta = .1$. Here too, it would of course be good to find a more meaningful SESOI, but I assume that the search would be even more difficult than for the main analysis. One important

issue needs to be fixed in any case: The authors seem to have followed my advice to determine equivalence using the CI. However, they report 95% CIs, not 90% CIs. Equivalence tests consist of two one-sided tests, each with a 5% alpha that is 'spent' on just one side. Because one cannot commit a type-I error on both sides simultaneously, the 5% of the two tests don't need to be corrected. This means that when using $\alpha = 5\%$, equivalence is determined with a 90% CI, not a 95% CI. Assuming that this is not merely a typo in the manuscript, the analysis should be updated, which may affect the conclusions.

6. Thank you for adopting this change.

7. Sticking with the preregistered two-sided tests in spite of a directed hypothesis is fine (as it is a conservative choice).

8. I appreciate the attempt to clarify the definition of 'robust'. To me, the revised sentence on p. 6, l. 216-217 still seems ambiguous though: 'We then seek to identify the mechanisms driving changes for only those outcomes that show significant and robust associations across both models and sub-items.' I think this could easily be fixed by changing the phrasing to '... that show significant associations which are robust across both models and sub-items.'

Finally, two small additional points:

- Equivalence test results on p. 8, l. 301-304: Here, a result is reported as *not* equivalent, yet with a p-value of $<.001$. Non-equivalent test results mean that the 90% CI crosses one of the bounds, yielding a p-value larger than .05 (and equivalence tests should always be reported with the larger one of the two p-values). I assume this p-value is a typo.

- Same paragraph, l. 306-308: The CI overlapping with one of the bounds is not sufficient for calling a result inconclusive. Inconclusive outcomes mean that neither the classic H_0 in the significance test against 0 nor the SESOI in the equivalence test can be rejected – i.e., a non-significant significance test *and* a non-significant equivalence test.

Thank you again for your efforts to address my comments. I think that the manuscript has notably improved. With the exception of my first comment here, all remaining points are minor and should be easily fixed.

Sincerely,

Anne Scheel

REVIEWER 8

#	Question/Comment	Response
1	I would like to thank the authors for their considerate engagement with my review comments. Several, though not all, of my previous points have now been resolved. I will go through them in order of the most recent rebuttal letter.	We appreciate your comments and suggestions for improving the SESOI.
2	1. My first comment concerned the justification for the SESOI of the equivalence tests for the central analysis (TL change between Wave 1 and Wave 2). In response, the authors changed their justification strategy and proposed two new SESOIs: the within-country and between-country variation (1 SD) of the outcome at Wave 1, set individually for TL, situation-specific norms, metanorms, and punishing frequency. I sincerely appreciate the authors' engagement with my arguments and their effort to provide a better solution. Unfortunately, though, I think that the new SESOI strategy is still not suitable (and may be less preferable than the previous version). First, the (within-country as well as between-country) variation of the outcome variable at Wave 1 is part of the effect size of the difference between Wave 1 and Wave 2. This means that the SESOI effectively depends on the effect size, which invalidates the equivalence test. For the test to be informative, the bound against which the effect is tested should be an independent criterion. Second, 1 standard deviation of the outcome variable is generally not a negligible effect size – it corresponds to a Cohen's d of 1. In many situations, finding that an effect is statistically smaller than $d = 1$ would be seen as not very informative. I understand this search for a SESOI may feel frustrating, especially because my lack of expertise in this research area means that I cannot provide a worked-out solution myself. The most pragmatic approach I can see at the moment is for the authors to investigate whether there are any mean differences in the data and findings of Wave 0 (Gelfand et al., 2011) that may serve as a reasonable benchmark. The guiding principle would be to find a contrast that is deemed theoretically inconsequential/uninteresting (or merely on the border to theoretically consequential/interesting). If no suitable empirical SESOI can be identified, I would consider going back to a default value such as a 'small effect' according to Cohen. I realise that this approach had been criticised by a previous reviewer and therefore changed, but if there are genuinely no other means of identifying a	Based on your arguments, we are convinced that using within and between country variance is unsuitable for setting the SESOI. Yet for the reasons that we describe in our prior reviewer response, using change in rank-ordering is also unsuitable. Moreover, other substantively motivated SESOIs are likewise unsuitable or unavailable. For instance, the reviewer mentions Gelfand et al. (2011)¹. However, the tightness-looseness data presented there (and also what are available) are transformed in a way that makes the raw values not comparable to our data. A similar problem, a lack of comparable data, holds more broadly for the other between wave changes as well as the mechanism changes (which is why we tried to use our own data for setting the SESOI). Based on this limitation, we revert to a benchmark-based approach, using Cohen's $d=0.1$ for the between wave analyses and $\beta = \pm 0.10$ for the mechanism analyses (p. 6). We note this situation explicitly in the text.

	SESOI that has meaning for the specific research area and target theory, I believe that defaults can be used as a last option. If this option must be used, I think it would be worth for the authors to note this situation explicitly, because understanding and building agreement about which effect sizes are meaningful is an important task for the entire research community in a field.	
3	2. The authors have implemented my request to clearly explain the justification for the SESOI in the text. My only remaining comment about this concerns the reporting of the results of the equivalence tests, in which tests results that are not equivalent are called 'practically meaningful'. I recommend against this phrasing: Whether or not an effect is seen as practically meaningful is not a matter of the outcome of an equivalence test per se, but depends on the SESOI. If the SESOI is not (practically) meaningful, the test result also doesn't provide such information. With this I just want to clarify that 'practically meaningful' is not a technical term like 'statistically significant' and should not be used mechanically.	We have changed the phrasing (p. 8).
4	3. The authors have also implemented my recommendation to set individual SESOIs for the tests of change in TL, situation-specific norms, metanorms, and punishing frequency. As I explained above, unfortunately I don't think that the new strategy for these tests is adequate. Ideally, my suggestion above could lead to finding meaningful SESOIs of each outcome variable. If this is impossible, a blanket SESOI will have to do (keeping the limitation of the apples vs oranges problem in mind).	See our response to comment #2.
5	4. Thank you for adding the descriptive statistics.	
6	5. The authors have now added equivalence tests for the 'mechanism' analyses, with a default SESOI of $\beta = .1$. Here too, it would of course be good to find a more meaningful SESOI, but I assume that the search would be even more difficult than for the main analysis. One important issue needs to be fixed in any case: The authors seem to have followed my advice to determine equivalence using the CI. However, they report 95% CIs, not 90% CIs. Equivalence tests consist of two one-sided tests, each with a 5% alpha that is 'spent' on just one side. Because one cannot commit a type-I error on both sides simultaneously, the 5% of the two tests don't need to be corrected. This means that when using $\alpha = 5\%$, equivalence is determined with a 90% CI, not a 95% CI. Assuming that this is not merely a typo in the manuscript, the	See our response to comment #2. We have updated our results and now use 90% CIs (pp. 14-15). Note that the values for the CIs sometimes do not change due to rounding.

	analysis should be updated, which may affect the conclusions.	
7	6. Thank you for adopting this change.	
8	7. Sticking with the preregistered two-sided tests in spite of a directed hypothesis is fine (as it is a conservative choice).	
9	8. I appreciate the attempt to clarify the definition of 'robust'. To me, the revised sentence on p. 6, l. 216-217 still seems ambiguous though: 'We then seek to identify the mechanisms driving changes for only those outcomes that show significant and robust associations across both models and sub-items.' I think this could easily be fixed by changing the phrasing to '... that show significant associations which are robust across both models and sub-items.'	We have adopted the reviewer's phrasing (p. 6).
10	Finally, two small additional points: - Equivalence test results on p. 8, l. 301-304: Here, a result is reported as *not* equivalent, yet with a p-value of <.001. Non-equivalent test results mean that the 90% CI crosses one of the bounds, yielding a p-value larger than .05 (and equivalence tests should always be reported with the larger one of the two p-values). I assume this p-value is a typo.	Thank you, corrected.
11	- Same paragraph, l. 306-308: The CI overlapping with one of the bounds is not sufficient for calling a result inconclusive. Inconclusive outcomes mean that neither the classic H0 in the significance test against 0 nor the SESOI in the equivalence test can be rejected – i.e., a non-significant significance test *and* a non-significant equivalence test.	We now specify that the results of the mechanisms analyses are both non-significant against zero and against the SESOI (p. 8).
12	Thank you again for your efforts to address my comments. I think that the manuscript has notably improved. With the exception of my first comment here, all remaining points are minor and should be easily fixed.	We also believe that the manuscript has improved due to your feedback and likewise appreciate your efforts and comments.

References

1. Gelfand, M. J. *et al.* Differences between tight and loose cultures: A 33-nation study. *Science* **332**, 1100–1104 (2011).

Reviewers' Comments:

Reviewer #8:

Remarks to the Author:

I thank the authors for their efforts to comply with my previous comments. Most of my concerns have been adequately addressed, with the exception of the following points:

1. The authors chose to revert back to defining the equivalence bounds with the benchmark approach (using a default value of a small effect size). As I explained in my last review, I am not opposed to this strategy. However, I was surprised that the authors claimed that the (admittedly not very concrete) alternative I suggested – exploring the data from Wave 0 (Gelfand et al., 2011) for mean differences that would be considered (just) negligible – was not feasible because these data "are transformed in a way that makes the raw values not comparable to our data". On p. 11 of the manuscript, the authors provide quite detailed analyses comparing changes in TL between Wave 0 and Wave 1, including on the country level. Unless I misunderstood something, these analyses seem to directly contradict the statement about Gelfand et al.'s data not being available in a useable form (which is now also included in the manuscript). I was additionally surprised about this since the first author of Gelfand et al. (2011) is also a co-author on this manuscript.

2. I previously noted that the results of equivalence tests should not be used to conclude that effects are or are not 'practically meaningful' unless the authors use equivalence bounds which reflect a true smallest effect size of interest. As the final approach is now to use a benchmark effect size instead, it should be clear that the test results are relative to these somewhat arbitrary bounds. The following phrases should be changed:

line 300: "implying that the differences are not substantively important given the SESOI we set"

line 302: "This implies that variation in hand washing is important."

line 310: " we find a small, albeit non-meaningful, decrease in tightness"

line 602: "the change is statistically equivalent (i.e., not of practical importance)."

If the authors wish to keep these references to importance or meaning, they have to explicitly argue that effect sizes smaller than $d = .1$ or $\beta = .1$ are not important/meaningful. I suggest instead to either remove these references or replace them with plain references to the equivalence bounds. For example, "not substantively important given the SESOI we set" can be replaced with "statistically smaller than the SESOI we set".

Minor comments:

3. In the descriptive statistics provided, two minimum and maximum values appear implausible to me:

line 467-468: Average COVID-19 fear, min = 3.42, max = 5.20 -> Neither value can be a valid response by one individual because the scale consists of only 3 items (i.e., neither "3.42" nor "5.2" can be the response of one person, unless the scale was continuous instead of a 6-point Likert scale).

line 484: min = 8.53, max = 42.65 -> These values seem similarly odd, unless the 0-100 scale was continuous?

Both cases could be explained by the descriptive stats referring to the country level rather than the individual level – apologies if this is the case and I misunderstood.

4. On lines 615-617, the verbal description of the equivalence test results for hand-washing norms change is wrong (the result is not equivalent and the effect estimate does not lie within the

bounds).

5. On line 621, the reported beta and CI for Government Stringency look like they might contain an error because the CI bounds are not symmetric around the effect estimate. This may be worth double-checking.

I have additionally been asked to comment on the authors' description of their power analyses and on their conclusions regarding the question whether non-significant results may or may not be due to low power. From my view, the description of the power analyses appears to be adequate. The authors draw two conclusions regarding power: on line 312-313 ("Importantly, the non-significant findings are due to the absence of substantial changes and not because of a lack of power.") and on line 348-349 ("Still, we do not have the power in the mechanisms analyses to detect small effects and we cannot entirely identify causality."). Both statements are supported by the presented evidence. The first statement could more clearly signal that it is referring to the analyses of mean differences between Wave 2 and Wave 1, though I think that this should be clear from the context in the respective paragraph.

Sincerely,

Anne Scheel

REVIEWER 8

#	Question/Comment	Response
1	The authors chose to revert back to defining the equivalence bounds with the benchmark approach (using a default value of a small effect size). As I explained in my last review, I am not opposed to this strategy. However, I was surprised that the authors claimed that the (admittedly not very concrete) alternative I suggested – exploring the data from Wave 0 (Gelfand et al., 2011) for mean differences that would be considered (just) negligible – was not feasible because these data "are transformed in a way that makes the raw values not comparable to our data". On p. 11 of the manuscript, the authors provide quite detailed analyses comparing changes in TL between Wave 0 and Wave 1, including on the country level. Unless I misunderstood something, these analyses seem to directly contradict the statement about Gelfand et al.'s data not being available in a useable form (which is now also included in the manuscript). I was additionally surprised about this since the first author of Gelfand et al. (2011) is also a co-author on this manuscript.	We recognise that this seems like an apparent contradiction. It is not. This is because the available data from Gelfand et al. (2011; Wave 0) can speak coherently to the question that we want to answer with the p. 11 analyses, which concern parallel trends in tightness-looseness pre-COVID 19, but cannot speak coherently to the question about mean changes in tightness-looseness, which is relevant for setting the SESOI. To explain this in more detail, the data reported in Gelfand et al. 2011 (Wave 0) are not available in their raw form and only after within-subject standardization. This procedure, which we also use (Wave 2) and so does Eriksson et al. (2021; Wave 1), aims to account for cultural tendencies for responding to surveys in certain ways (e.g. systematically more/less extreme), and is done by calculating the mean for each person's responses to a set of questions that do not concern tightness-looseness in the survey and then subtracting that mean from each tightness-looseness item. Since our survey, and that conducted by Gelfand et al. 2011, differ in the non-tightness-looseness items used for standardization, there will be systematic differences between the adjusted values of the countries reported in Gelfand et al. 2011 and the values that we find. This can be directly seen by comparing the country scores reported in Gelfand et al. 2011 (Table 1) and our country scores (Table S1). Because of these differences, we cannot meaningfully compare the changes in absolute values between Wave 0 and Wave 1, preventing us from identifying a SESOI based on average changes in tightness-looseness of countries. Nevertheless, we are able to use the Wave 0 data for another purpose: to check for parallel trends in the tightness-looseness of countries pre-COVID. Specifically, in the analysis mentioned on p. 11 (also Supplementary material Figure S7, Table S14), we check whether the countries that later (between Wave 1 and 2) are more/less affected by COVID-19 already systematically differ in their trends beforehand. To check for

		this, we mean centre tightness-looseness in all three waves and standardize the scores (i.e. divide by the SD). This allows us to identify if countries that were tight in Wave 0, relative to other countries in Wave 0, remain tight in Wave 1 relative to the other countries in Wave 1 or vice versa and to detect any meaningful trends. This is sufficient to check for parallel trends since if the assumption is violated, countries that were less/more affected by COVID would systematically deviate in their relative tightness-looseness across the waves. Yet since we standardise and mean centre tightness-looseness, average change in tightness looseness between Wave 0 and 1 cannot be calculated meaningfully (i.e. overall average change is 0). There are two further points worth highlighting concerning this comment. First, even though the absolute values differ between Wave 0 and Wave 1, and we do not have access to the raw data from Wave 0 (while Gelfand is a co-author on this manuscript those data were collected over 20 years ago), the Wave 0 country-level values correlate very strongly with both Wave 1 ($r=0.89$) and Wave 2 ($r=0.88$) reassuring us that the Wave 0 data are meaningful but adjusted according to the within-subject standardization procedure that they describe (Supplementary Materials, p. 2). Second, because we use exactly the same items to standardize tightness-looseness as Eriksson et al. (2021; Wave 1), we are able to compare meaningfully Wave 1 to Wave 2. Moreover, comparing unstandardized tightness-looseness scores across waves 1 and 2 leads to the same findings. We realise that these details are sparse in the manuscript (due to space limitations) but we have added an explanation in the Supplementary Materials (p. 30).
2	I previously noted that the results of equivalence tests should not be used to conclude that effects are or are not 'practically meaningful' unless the authors use equivalence bounds which reflect a true smallest effect size of interest. As the final approach is now to use a benchmark effect size instead, it should be clear	We have adjusted the phrasing to that suggested by the reviewer.

	that the test results are relative to these somewhat arbitrary bounds. The following phrases should be changed: line 300: "implying that the differences are not substantively important given the SESOI we set" line 302: "This implies that variation in hand washing is important. line 310: " we find a small, albeit non-meaningful, decrease in tightness" line 602: "the change is statistically equivalent (i.e., not of practical importance)." If the authors wish to keep these references to importance or meaning, they have to explicitly argue that effect sizes smaller than $d = .1$ or $\beta = .1$ are not important/meaningful. I suggest instead to either remove these references or replace them with plain references to the equivalence bounds. For example, "not substantively important given the SESOI we set" can be replaced with "statistically smaller than the SESOI we set".	
3	In the descriptive statistics provided, two minimum and maximum values appear implausible to me: line 467-468: Average COVID-19 fear, min = 3.42, max = 5.20 → Neither value can be a valid response by one individual because the scale consists of only 3 items (i.e., neither "3.42" nor "5.2" can be the response of one person, unless the scale was continuous instead of a 6-point Likert scale). line 484: min = 8.53, max = 42.65 → These values seem similarly odd, unless the 0-100 scale was continuous? Both cases could be explained by the descriptive stats referring to the country level rather than the individual level – apologies if this is the case and I misunderstood.	As explained in the manuscript (lines 472-476), the variable Fear of COVID-19 is the average of three items. The minimum and maximum reported refer to the average of such three items across all individuals in our dataset and hence it can include decimal values. Similarly, for COVID-19 prevalence.
4	On lines 615-617, the verbal description of the equivalence test results for hand-washing norms change is wrong (the result is not equivalent and the effect estimate does not lie within the bounds).	Thank you for noticing this error. We have corrected it.
5	On line 621, the reported beta and CI for Government Stringency look like they might contain an error	Done; there was a negative sign incorrectly included for the beta. The CI bounds are correct however.

	because the CI bounds are not symmetric around the effect estimate. This may be worth double-checking.	
--	--	--

References

Gelfand, M. J. *et al.* Differences between tight and loose cultures: A 33-nation study. *Science* **332**, 1100–1104 (2011).